# FOR-PROMPTING: FROM OBJECTION TO REVISION VIA AN ASYMMETRIC PROMPTING PROTOCOL

## ABSTRACT

Reasoning protocols such as Chain of Thought (CoT) and Tree of Thought (ToT) organize internal deliberation but lack an explicit mechanism for external questioning that elicits self-revision. We present FOR-Prompting (From Objection to Revision Prompting), an asymmetric protocol where a Defender proposes an answer, an Debater (Questioner) raises question-style objections with no direct fixes, and a Host optionally synthesizes the final output. Across GSM8K, FOR-Prompting matches the accuracy of CoT and consistently improves over single-prompting when evaluated under identical model backbones. On small-scale open-source models (e.g., LLaMA-3.2-1B), FOR-Prompting yields substantial gains over direct prompting and performs comparably to lightweight reasoning baselines, highlighting its promise for low-resource and on-device settings. Cross-model role-swapping further shows that performance is primarily determined by the Defender, enabling small models to act effectively as Questioners. Beyond structured math tasks, FOR-Prompting supports refinement in open-ended and multi-stage tasks: qualitative analysis shows improved exploration, coverage, and specificity, and a blind human preference study found that participants preferred FOR-Prompting outputs over strong LLM baselines in an itinerary-planning scenario. The protocol is model-agnostic and operates purely through role-structured prompting, requiring no training, access to model internals, or symmetrically strong agents. FOR-Prompting therefore enables scalable study of objection-driven reasoning and offers a practical mechanism for automated iterative refinement across both hosted and local LLMs.

## 1 INTRODUCTION

Large language models (LLMs) increasingly solve complex tasks by following prompting strategies that structure their intermediate reasoning. Within this line of work, Chain-of-Thought (CoT) (Wei et al., 2022) encourages linear stepwise explication, Tree-of-Thought (ToT) (Yao et al., 2023a) organizes branched exploration, Self-Consistency (Wang et al., 2023b) aggregates multiple independent chains, and self critique aims to repair an initial draft through reflective prompts. More recently, graph structured prompting such as Graph-of-Thoughts (GoT) (Besta et al., 2024) extends beyond trees by exploring and scoring a broader space of partial ideas. Taken together, these single agent strategies have advanced transparency and control. Yet these protocols operate primarily within a single reasoner, lacking an explicit mechanism for external questioning that elicits self-revision without supplying solutions. In many settings, such as mathematical word problems, planning, analysis, or explanation, what improves an answer is not another answer, but a well-posed question that surfaces gaps, unstated assumptions, or overlooked constraints.

A parallel literature explores multi agent prompting. Debate style systems pit agents against each other (Du et al., 2024; Park & Seo, 2025) , committee and ensemble schemes solicit multiple answers and then vote or justify, reviewer and reviser workflows attach a critic that proposes edits, tutor or coach patterns offer hints or worked substeps, and cross examination has agents interrogate one another before an aggregator selects a final output. These designs introduce valuable external pressure (i.e., external questioning applied by a separate model rather than self-reflection), yet they typically inject external thinking and sometimes competing solutions into the trace. As a result, error detection is often confounded with answer replacement, provenance becomes diluted because multiple authors shape the content, and evaluation grows complicated because improvements may stem

from the external agent rather than from the solver's revised reasoning. Iteration depth is not always controlled and outcome level evaluation is often mixed with process supervision, which makes it hard to isolate the effect of questioning itself. These gaps motivate an alternative that keeps the benefits of external pressure while avoiding external solution content.

We draw on a widely used human-in-the-loop (HITL) practice (Wu et al., 2022). Human reviewers rarely fix an LLM's response directly. Instead, they improve the quality of the LLM's responses or request more detailed explanations by posing questions. These questions reveal missing assumptions, inconsistent steps, ambiguous concepts, or infeasible constraints. We introduce **FOR-Prompting**, short for From Objection to Revision Prompting. A Defender proposes an answer. A Questioner then asks question-style challenges rather than supplying fixes, with the aim of inducing the Defender's rethinking and revision. Here objection is used in a broad questioning sense that includes clarifying questions, constraint checks, and counterexample or counterfactual probes. By separating roles so that the Questioner asks and the Defender reasons and revises, FOR-Prompting preserves a single accountable line of thought and turns questioning into a first class mechanism for revision. Conceptually, this can be viewed as an automated HITL-like question loop, where the human-like interrogative pressure is delivered by an agent, mimicking the questioning behavior humans typically use when iteratively refining LLM outputs rather than involving real human users.

Methodologically the protocol is open ended by design. It can iterate without a fixed upper limit or stop under a user specified rule such as a utility threshold, a convergence signal, a time or cost budget, or a planner defined milestone. This flexibility is useful for brainstorming and refinement heavy work including multi step planning, scenario exploration, and requirement elicitation, where additional cycles of questioning often surface new constraints or alternatives. For controlled evaluation in this paper we instantiate an external only setting. A Questioner poses clarifying and adversarial questions and a Defender revises the solution in response. No self reflection step is inserted. The Questioner never supplies direct fixes and the Defender remains the sole author of revisions. We cap the number of rounds with a small constant that is kept fixed within each experiment and reported with the results.

In our experiments, we conducted four case studies to validate the effectiveness of FOR-Prompting: (1) Benchmark evaluation on GSM8K (Cobbe et al., 2021): using GPT-4o, we compared FOR-Prompting against single-prompt, CoT, Self-Ask, and Self-Consistency baselines under identical settings. FOR-Prompting matched the performance of CoT and Self-Ask, and remained close to Self-Consistency, while consistently outperforming the single-prompt baseline. These results confirm that external question-driven refinement can be as effective as established reasoning scaffolds. (2) Small-scale model evaluation: on LLaMA-3.2:1B, we compared single-prompt, CoT, and two FOR-Prompting variants (with and without Host aggregation). Both FOR-Prompting variants substantially outperformed the single-prompt baseline and achieved accuracy comparable to CoT. Host aggregation provided limited benefit on small models due to summarization errors, highlighting that the gain primarily comes from the external-questioning mechanism. (3) Cross-model role swapping: we show that using a small model as the Questioner and a stronger model as the Defender preserves high performance while significantly reducing large-model token usage. This asymmetric structure differs from self-ask–style methods and provides a cost-efficient strategy for hybrid pipelines. (4) Open-ended tasks: FOR-Prompting produced more complete and actionable plans than baselines on complex itinerary-generation tasks. A blind human preference study with 77 participants shows that 74% preferred the FOR-Prompting output over those from two strong frontier LLM systems.

In summary, we present FOR-Prompting as an automated form of HITL-like questioning that preserves a single authorial line of thought while benefiting from external pressure. By treating questions as the sole external input and without requiring direct fixes, FOR-Prompting enables controlled study of questioning itself rather than a mixture of questioning and external reasoning, thereby improving interpretability, accountability, and practical utility.

## 2 CONTRIBUTIONS

This work makes three primary contributions. (1) Conceptual novelty: We introduce FOR-Prompting, the first prompting protocol that formalizes questioning-rather than answer substitution-as the exclusive form of external intervention. By separating questioning from revision, FOR-Prompting preserves a single accountable reasoning chain while leveraging external pressure to elicit self-revision. (2) Protocol design: We instantiate FOR-Prompting as a lightweight, role-based inter-

action loop between a Defender and a Questioner, enforcing that objections take the form of questions only, which enables systematic study of questioning as a mechanism for improving reasoning. (3) Empirical validation: We validate FOR-Prompting on both commercial and open-source LLMs. On GSM8K, FOR-Prompting achieves higher accuracy than the baseline and matches the same level of the performance of CoT. When tested on small-scale open-source models, FOR-Prompting yields substantial accuracy gains over the baseline. In addition, we provide illustrative examples showing that FOR-Prompting can effectively correct errors in LLM outputs and refine or expand open-ended tasks, serving as a viable alternative to direct human supervision or intervention. In open-ended itinerary-planning scenarios, FOR-Prompting consistently generated more complete, realistic, and actionable plans than strong baseline models, as also reflected in human preference evaluations.

## 3 RELATED WORK

**Reasoning and planning in single-agent prompting.**  LLMs have popularized explicit intermediate reasoning via CoT (Wei et al., 2022), which improves performance on structured tasks, while Self-Consistency (Wang et al., 2023b) aggregates diverse reasoning paths to reduce brittle single-trace errors.  Beyond linear chains, ToT (Yao et al., 2023a) explores alternative branches with look-ahead and backtracking, and GoT (Besta et al., 2024) generalizes this idea to non-linear structures. Complementary scaffolds strengthen problem decomposition and execution. Planning-based prompting includes *least-to-most* (Zhou et al., 2023) and plan-then-solve (Wang et al., 2023a). Tool-use methods such as PAL (Gao et al., 2023) and PoT (Chen et al., 2023) delegate subproblems to programs. ReAct (Yao et al., 2023b) interleaves reasoning with external actions to ground steps in evidence. Self-ask Press et al. (2023) promotes reasoning by having the model ask itself follow-up questions and then answer them within a single-model loop. These approaches increase transparency and controllability, yet they primarily operate within a single reasoner and therefore provide only limited avenues for externally driven improvement through questioning.

**Self-reflection and verification mechanisms.**  A parallel line of work adds self-assessment and correction. Self-Refine iteratively revises outputs using model-generated feedback (Madaan et al., 2023), and Reflexion maintains episodic reflective memory to guide adaptation (Shinn et al., 2023). Constitutional AI encodes explicit normative constraints to regulate outputs (Bai et al., 2022). Chain-of-Verification validates intermediate claims before targeted revision (Cobbe et al., 2021; Dhuliawala et al., 2024). SelfCheckGPT further automates verification through sampling and consistency checking (Manakul et al., 2023).  Recent work also documents limits of reflection, including an early-stop effect where gains plateau after a few iterations. DORA sustains progress with dynamic optimization prompts (Li et al., 2025), and Agent-Pro uses policy-level reflection and optimization to evolve agent behaviors over time (Zhang et al., 2024a). Beyond single-agent reflection, reflective multi-agent collaboration distributes critique and self-reflection across multiple agents to improve robustness and adaptability on complex tasks (Bo et al., 2025). These studies underscore the value of feedback and verification, while also revealing that internal reflection alone may be insufficient when external perspectives are needed to challenge early commitments.

**Multi-agent prompting for reasoning and collaboration.**  Multi-agent systems (MAS) (Dorri et al., 2018; He et al., 2025; Sun et al., 2025; Maldonado et al., 2024) frame reasoning and collaboration as interactions among specialized agents rather than a single model acting alone. By distributing roles, perspectives, and responsibilities, MAS approaches target robustness, explainability, and creativity (Naik et al., 2025; Epperson et al., 2025; Zhang et al., 2024b). Within this paradigm, debate-style frameworks position agents with opposing views to contest one another, which can surface errors through adversarial questioning and improve factual accuracy and robustness (Du et al., 2024). Park & Seo (2025) report that multi-agent debate aligns more closely with human evaluators in educational assessment than single-agent baselines. Purely agent–agent debate such as MAD shows reductions in hallucination and gains in accuracy (Du et al., 2024), while reflective multi-agent collaboration distributes critique and synthesis across agents to support long-horizon tasks (Bo et al., 2025). At the same time, recent findings warn that multi-agent settings can suffer social influence effects that reduce robustness under peer pressure (Song et al., 2025).

**Role specialization in multi-agent architectures.**  Beyond symmetric debate, role-based designs coordinate complementary skills to tackle complex tasks. CAMEL demonstrates that role-playing agents can sustain goal-directed exchanges with minimal human steering (Li et al., 2023). Auto-

Gen provides an extensible infrastructure for assistant–critic and team-based interactions (Wu et al., 2024). Debate has also been adapted to encourage divergent reasoning paths rather than quick consensus (Liang et al., 2024). In software engineering, ChatDev and MetaGPT orchestrate requirements, design, coding, and testing through division of labor and iterative review, which scales solution quality in practice (Qian et al., 2024; Hong et al., 2024). Despite these advances, many systems converge episodically through voting or consensus, which relaxes sustained pressure to interrogate a reasoning chain, and many prioritize factual question answering over open-ended planning and creative refinement, which limits their utility in tasks that demand progressive elaboration.

**Dialogue-based deliberation and automated reflection.** Multi-agent dialogue extends beyond adversarial debate to simulate deliberation, reflection, and negotiation. Zhang et al. (2024b) design a generative MAS that helps users burst filter bubbles by engaging diverse perspectives through gamified interactions. Ashkinaze et al. (2025) guide LLMs with simulated social ensembles that follow procedures from deliberative democracy. Divekar et al. (2020) develop a human–multi-agent negotiation platform in immersive environments, highlighting opportunities and challenges of multiparty dialogue. Dialogue frameworks also operationalize draft–critique–revise loops. AutoGen's assistant–critic pattern (Wu et al., 2024), debate with adjudication (Liang et al., 2024), and LLM-as-judge protocols for scalable evaluation (Zheng et al., 2023) are representative. However, most of these systems target single-answer accuracy under bounded contexts and often embed hints, templates, or edits, which makes it difficult to isolate the effect of questioning itself.

**Positioning.** The studies above either keep all reasoning inside one agent or allow external collaborators to introduce hints, edits, or competing solutions. In contrast, our setting focuses on externally driven improvement through questions alone. We automate a HITL-like practice in which targeted questions expose gaps, assumptions, and constraints, while the answering agent remains solely responsible for the revision. Unlike Self-Ask, which requires a single model to generate and answer its own follow-ups, our method assigns answering and questioning to different agents. This prevents self-confirmation bias, introduces genuinely external interrogative pressure, and preserves a single accountable reasoning trajectory without entangling solution generation with self-evaluation. Also, unlike multi-agent debate, the Questioner in our framework does not generate competing solutions or attempt to win an argument; it contributes only question-style challenges, making the interaction asymmetric and refinement-oriented rather than adversarial. By treating questions as the only currency of intervention, our approach preserves a single accountable line of thought, avoids confounds from externally supplied reasoning, and complements debate, review, and tutoring styles with a mechanism for persistent and structured questioning that supports open-ended planning and iterative refinement.

## 4 METHODOLOGY

We introduce FOR-Prompting, a multi-agent, questioning-driven prompting protocol that structures inference as an iterative question/challenge-revision loop. Given a question $Q$ and a round budget $N$, the protocol coordinates three roles: a *Defender* that proposes and iteratively revises solutions, a *Debater* that injects external objections/questions without revealing solutions, and an optional *Host* that synthesizes the final answer from the full interaction history.

The process unfolds as follows. The Defender first produces an initial answer $A_0$ to $Q$. The Debater then analyzes $A_0$ and asks a targeted question $Q_1$, prompting the Defender to revise its solution to $A_1$ conditioned on $Q$ and $Q_1$. Repeating this cycle for $N$ rounds yields a trajectory

$$\{A_0, Q_1, A_1, \ldots, Q_N, A_N\}.$$

Finally, the Host aggregates the dialogue trajectory into the final answer $A^*$.

Formally, for $r = 1, \ldots, N$,

$$A_0 = \text{Defender}(Q),$$
$$O_r = \text{Debater}(A_{r-1}),$$
$$A_r = \text{Defender}(Q, O_1, \ldots, O_r),$$
$$A^* = \text{Host}(Q, \{A_0, O_1, A_1, \ldots, O_N, A_N\}).$$

**Role Responsibilities.** The three roles follow strict divisions of responsibility: *Defender* proposes and revises solutions under questioning; *Debater* issues targeted questions only (no alternative solutions), focusing on logical gaps, hidden assumptions, counterexamples, and robustness; *Host* integrates the multi-round trace, ensures resolution of concerns, and produces the final answer $A^*$.

**Algorithm 1** FOR-Prompting Protocol (abstracted pseudocode)

---
**Require:** Question $Q$, round budget $N$
1: $A_0 \leftarrow \mathsf{Defender}(Q)$
2: **for** $r = 1$ to $N$ **do**
3:     $O_r \leftarrow \mathsf{Debater}(A_{r-1})$
4:     $A_r \leftarrow \mathsf{Defender}(Q, O_1, \ldots, O_r)$
5: **end for**
6: $A^* \leftarrow \mathsf{Host}(Q, \{A_0, O_1, A_1, \ldots, O_N, A_N\})$
7: **return** $A^*$

---

These responsibilities are enforced through role-specific prompts that constrain outputs to questions, revisions, or synthesis, while leaving surface form flexible. The aim is not to prescribe a fixed template but to enforce a mechanism of externally driven questioning and revision.

**Protocol Representation.** To ensure transparency and analyzability, each run records a structured dialogue history with role tags and message types, which guides Host synthesis and enables subsequent process-level analysis.

Algorithm 1 presents an abstract pseudocode version of FOR-Prompting, capturing the essential dynamics of questioning and revision while omitting implementation-specific details. Figure 1 complements this abstraction with a flowchart illustrating the iterative structure.

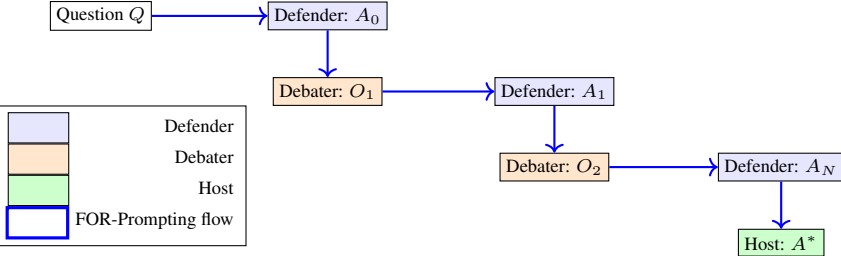

Figure 1: FOR-Prompting protocol flow (external questions only). The Debater raises questions, the Defender revises answers, and the Host synthesizes the final output. *Host could be optional.

## 5 EXPERIMENTAL SETUP AND EVALUATION

### 5.1 TESTING ON GSM8K

#### 5.1.1 CASE STUDY 1: LARGE LANGUAGE MODEL (GPT-4O) ON GSM8K

**Datasets.** We first evaluate on GSM8K (Cobbe et al., 2021), a elementary and middle school math word problems benchmark. We evaluate on the public test set and, for local-model experiments, report results on a uniformly sampled subset of 500 test items with a fixed seed. Each method produces a free-form solution; no prompt length cap is imposed. All runs persist the full dialogue for auditability, while metrics are computed from the final solution produced for each item.

**Accuracy evaluation.** For each model response, we deterministically extract the final numeric prediction and compare it against the ground-truth solution. GPT-4.1 (temperature 0) is used only as a preliminary verifier to assist rapid inspection, but the final correctness labels are manually checked to eliminate judge-model errors. All extraction, verification, and scoring procedures are applied uniformly across the 500 sampled GSM8K items.

Table 1: GSM8K accuracy across prompting methods using `gpt-4o`.

| Method | Accuracy |
|---|---|
| Single-prompt | 0.92 |
| Chain-of-Thought (CoT) | 0.94 |
| Self-ask | 0.94 |
| Self-consistency | 0.95 |
| FOR-Prompting (Host=True) | 0.94 |
| FOR-Prompting (Host=False) | 0.94 |

**APIs and determinism.** All methods are executed through the same programmatic framework with temperature 0 for reproducibility. Unless otherwise noted, all model-generated responses across experiments use OpenAI's `gpt-4o` through the official OpenAI API. This ensures that differences in performance arise from the prompting protocols rather than backend model variations. All LLM-as-a-judge evaluations are conducted using OpenAI's GPT-4.1.

**Results.** Table 1 reports GSM8K accuracy across prompting methods under a unified `gpt-4o` backend. The single-prompt baseline achieves an accuracy of 0.92. All structured prompting methods yield modest improvements over this baseline: CoT and self-ask both reach 0.94, self-consistency achieves 0.95, and FOR-Prompting attains 0.94 in both its Host and no-Host variants. Overall, these results indicate that structured reasoning generally provides a small but consistent benefit over single-shot prompting, with all advanced prompting strategies exhibiting comparable performance.

### 5.1.2 CASE STUDY 2: SMALL-SCALE MODEL (LLAMA-3.2:1B) ON GSM8K

**Setting.** We evaluate an open-source 1B-parameter model (LLaMA-3.2:1B[1][2]) on the 1,319-item GSM8K test set to examine whether lightweight interaction structures can compensate for limited parametric capacity. We compare four prompting strategies: (i) **Single-prompt**: a direct answer without explicit reasoning. (ii) **CoT**: prompting the model to generate step-by-step reasoning before an answer. (iii) **FOR-Prompting (Host-True)**: the standard protocol in which a Host synthesizes the final answer. (iv) **FOR-Prompting (Host-False)**: the same protocol without Host synthesis; the Defender's final revision is returned directly.

In both FOR-Prompting variants, the Debater provides only question-style challenges and never supplies solutions; the Defender re-answers the original question each round to avoid objective drift. Temperature and decoding settings are held constant across all methods.

**Results.** Table 2 reports accuracy on the GSM8K. The **Single-prompt** baseline yields only **7%** accuracy, confirming the difficulty of multi-step reasoning for a 1B-parameter model. Adding **CoT** substantially improves performance to **23%**, showing that even very small models benefit from explicit reasoning scaffolds.

Table 2: Accuracy of LLaMA-3.2:1B on GSM8K under different strategies.

| Method | Accuracy |
|---|---|
| Single-prompt | 0.07 |
| Chain-of-Thought (CoT) | 0.23 |
| FOR-Prompting (Host-True) | 0.19 |
| FOR-Prompting (Host-False) | 0.23 |

FOR-Prompting achieves competitive performance relative to CoT. The **Host-False** variant reaches **23%**, matching CoT, while the **Host-True** version obtains **19%**. Importantly, both FOR-Prompting variants substantially outperform the single-prompt baseline, demonstrating that the external-questioning mechanism remains effective even under severe model capacity constraints. In particular, despite the model's limited size, a single round of structured questioning enables the model to surface missing constraints and correct earlier reasoning gaps, capabilities that the single-prompt baseline rarely exhibits.

Although the Host role is primarily intended to provide a clearer final presentation rather than improve reasoning quality, the results indicate that its impact on small-model accuracy is limited. For a 1B model, the additional synthesis step can occasionally introduce errors, such as dropping constraints or mis-merging intermediate reasoning. Moreover, the extra consolidation stage can amplify the model's inherent noise, since each additional generation step compounds earlier uncertainties. This explains why **Host-False** slightly outperforms **Host-True**: lightweight models tend to be more reliable when returning their final reasoning trace directly rather than performing an extra consolidation step.

Overall, while the absolute accuracy remains far below that of large-scale models, the consistent gain over the single-prompt baseline, more than doubling accuracy, shows that structured prompting strategies, including both CoT and FOR-Prompting, meaningfully amplify reasoning ability in small models.

### 5.1.3 CROSS-MODEL ROLE SWAPPING EXPERIMENTS

To examine whether FOR-Prompting remains robust when different models assume the Defender and Debater roles, we conducted a cross-model role swapping evaluation using GPT-4o and LLaMA-3.2:1B.

Figure 2: Cross-Model Role Swapping Results

| Defender Model | Debater Model | Method | Accuracy |
|---|---|---|---|
| GPT-4o | LLaMA-3.2:1B | FOR-Prompting | 0.93 |
| LLaMA-3.2:1B | GPT-4o | FOR-Prompting | 0.21 |

This setting tests the asymmetric nature of the protocol by observing how performance changes when model capacities are permuted across roles.

---

[1] https://ollama.com/library/llama3.2:1b

[2] https://ai.meta.com/blog/llama-3-2-connect-2024-vision-edge-mobile-devices/

Table 2 summarizes the results. When GPT-4o served as the Defender and LLaMA-3.2:1B acted as the Debater, FOR-Prompting achieved an accuracy of 0.932. This suggests that the Debater's role, primarily generating external interrogative pressure, does not require high model capacity to be effective. In contrast, when the roles were reversed, and LLaMA-3.2:1B served as the Defender while GPT-4o served as the Debater, accuracy dropped sharply to 0.21. This indicates that the Defender role is substantially more sensitive to model expressiveness, since it must propose candidate answers and integrate external challenges into the final reasoning.

These results reinforce our central claim that FOR-Prompting is an *asymmetric prompting protocol*. The protocol does not require symmetry in model capability: a small model can serve as an effective Debater, but strong performance depends critically on the Defender being sufficiently capable.

## 5.2 CASE STUDY 3: CORRECTING MISTAKES THROUGH EXTERNAL QUESTIONS

In this case study, we use a currently popular commercial model and apply the FOR-Prompting method to address problems where these models would otherwise produce incorrect answers. The results show that FOR-Prompting can correct erroneous responses without human supervision or external tool assistance.

Specifically, we present a tricky question, "How many rs in starrtrabbbery" (this is the only prompt we input to LLM), which is often incorrectly answered by LLMs because of their tendency to miscount characters in strings due to tokenization and attention limitations. We used GPT-4.1 to answer this question. Under the single-prompt setting, we obtained an incorrect response ("There are 4 'r's in 'starrtrabbbery.'"). However, within the FOR-Prompting framework, the debater raised additional questions that guided the defender back on track, resulting in the correct answer ("Therefore, there are 5 'r's in 'starrtrabbbery.'"), as shown in Figure 3.

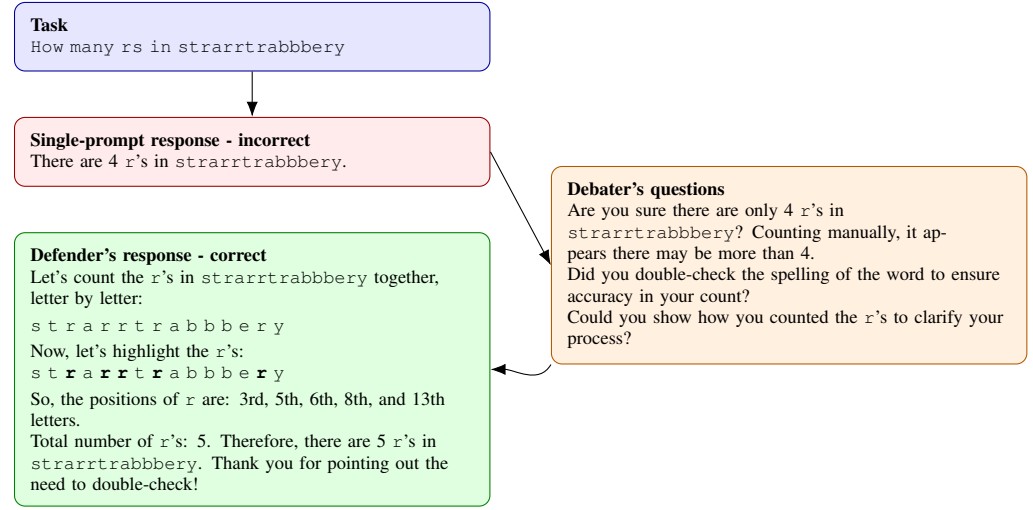

Figure 3: A detailed example of FOR-Prompting with initial task, incorrect single-prompt answer, debater's questions, and corrected defender response.

## 5.3 CASE STUDY 4: REFINING OPEN-ENDED TASK THROUGH EXTERNAL QUESTIONS

We compared three approaches, ChatGPT 5 Thinking (Extended thinking) [3], GPT-4.1 (single-prompt), and FOR-Prompting (GPT-4.1-based), on the open-ended task "*Create a 5-day itinerary for a trip to Rio de Janeiro in April.*" This kind of task, an open-ended itinerary-planning and content-generation task, involves integrating multi-objective constraints, step-by-step reasoning, and fact-checking. Therefore, evaluating models on such tasks enables a comprehensive assessment of their overall capabilities and practical value for real-world decision support and alignment with user preferences.

Qualitatively, FOR-Prompting yielded more robust, field-ready itineraries than the other two approaches. For example, when the Debater asked "*what to do if tickets to Christ the Redeemer were*

[3] https://openai.com/index/introducing-gpt-5/

*unavailable*", the revision added clear alternatives and timing guidance; when the Debater questioned "*whether combining Sugarloaf and Lapa in one day was realistic*", the plan incorporated time buffers and fallbacks. Table 3 summarizes the relative breadth of practical information across the three methods. From this we can see that FOR-Prompting considers far more factors in its plan than ChatGPT 5 and the single-prompt baseline. Although we can prompt an LLM with the relevant factors and have it add the missing content-something models and tools, especially a stronger model like ChatGPT 5, can readily do to supplement, expand, and refine-the process typically relies on human expertise (manual prompting) rather than being fully automated as in FOR-Prompting. The complete FOR-Prompting content appears in Appendix A.3.

| Factor | ChatGPT 5 (Extended thinking) | GPT-4.1 (Single-prompt) | FOR-Prompting (GPT-4.1-based) |
|---|---|---|---|
| Weather | ✓ | ✓ | ✓ |
| Packing/Attire | ✓ | ✓ | ✓ |
| Reservations/Tickets | ✓ | - | ✓ |
| Transportation | ✓ | - | ✓ |
| Accessibility | - | - | ✓ |
| Safety | ✓ | ✓ | ✓ |
| Rainy-day alternatives | ✓ | - | ✓ |
| Alternatives/Branching | ✓ | - | ✓ |
| Nightlife/Music | ✓ | ✓ | ✓ |
| Food/Restaurants | ✓ | ✓ | ✓ |
| Markets/Shopping | - | ✓ | ✓ |
| Language/Comms | - | - | ✓ |
| Offline/Navigation | - | - | ✓ |
| Luggage Storage | - | - | ✓ |
| Emergency Info | - | - | ✓ |
| Holidays/Hours | - | - | ✓ |
| Photography/Viewpoints | ✓ | ✓ | ✓ |
| Signature Experiences | ✓ | - | ✓ |
| Family-friendly | - | - | ✓ |
| Health/Water | - | ✓ | ✓ |

Table 3: Practical Info & Coverage Comparison

**Human preference evaluation.** To examine whether the benefits of iterative external questioning generalize beyond a single prompt, we conducted a separate human preference study using an additional open-ended planning task: "*Create a 7-day travel itinerary for Barcelona.*" This task belongs to the same class of multi-constraint itinerary-planning problems as the Rio example but offers a different geographic and structural context, enabling a broader test of generality.

We recruited 77 participants via social media platforms. Each participant completed a single-choice question after reviewing three anonymized outputs produced by: (1) FOR-Prompting (GPT-4.1-based), (2) GPT-4.1 (single-prompt), and (3) a widely used frontier LLM system. Participants were asked: "Which of the following outputs do you prefer?" Results show that **74% (57/77)** of participants preferred the FOR-Prompting itinerary. Participants highlighted its greater completeness, more realistic daily pacing, clearer contingency planning, and stronger practical utility. These findings indicate that iterative external questioning enhances open-ended planning performance not only in the Rio case study but also in a distinct and independently evaluated scenario, reinforcing the broader applicability of FOR-Prompting for real-world decision support.

## 5.4 CASE STUDY 5: MULTI-STAGE TASK EXTENSION OF FOR-PROMPTING

Given the growing demand for multi-stage agentic tasks, we added a small pilot study to illustrate the applicability of FOR-Prompting in dynamic, multi-stage settings. As shown in Figure 4, after completing Task 1 we introduced new conditions or new knowledge. These additions reflect realistic sources of task updates, such as temporal changes, real-world constraints, emerging requirements, or subsequent tasks that depend on prior outputs.

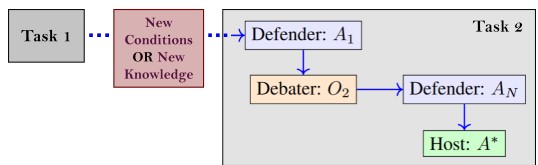

Figure 4: Multi-stage extension of FOR-Prompting. New conditions or new knowledge are incorporated after Task 1, enabling iterative refinement toward the Task 2 solution.

We reused the travel-planning scenario from Case Study 4 and treated its final response as the completed output of Task 1. We then assumed that on Day 4 of the trip, additional travelers would join, forming a six-person group for that day's activities. This new group-travel condition was added

to the task specification, and FOR-Prompting was run for three additional iterative rounds, which constitute Task 2.

The pilot results show that FOR-Prompting incorporates newly added conditions rather than treating each turn as an isolated optimization. Beyond general considerations such as distance or dining options, the notion of group travel was explicitly integrated into the objections raised by the Debater. Example inquiries include: *"For group logistics, did you consider the potential for traffic delays or the availability of large Ubers or private vans during peak times?"* and *"For artisanal shops and farmers' markets, will a group of six moving together disrupt the flow or experience?"* These questions capture factors that only emerge once group size becomes a relevant constraint.

New conditions or new knowledge can originate from multiple sources. Users may directly provide new inputs that modify earlier assumptions, or external APIs can introduce real-time updates, such as local news feeds, delayed transportation alerts, or weather forecasts. This mechanism makes FOR-Prompting particularly suitable for tasks that require frequent adjustment or multi-stage processing based on ongoing events or new data. Examples include strategy revision, policy adjustment, and dynamic analysis tasks in domains such as healthcare, finance, education, and other environments where conditions evolve continuously.

These observations point to FOR-Prompting's potential to handle dynamic multi-stage updates and adapt to evolving task requirements. Full outputs are provided in Appendix A.4.

## 6 DISCUSSION

### 6.1 HUMAN-INSPIRED EXTERNAL QUESTIONING

FOR-Prompting draws inspiration from human collaborative reasoning, where interlocutors improve solutions by asking clarifying and adversarial questions rather than by directly proposing alternatives. However, while the approach is inspired by human collaboration, FOR-Prompting deliberately removes the need for human-in-the-loop prompting during problem solving. Human oversight is often bounded by uneven domain knowledge and experiential blind spots, leading to under-specified goals or misread constraints (Zamfirescu-Pereira et al., 2023; Zhang et al., 2025a). Moreover, even expert users may struggle to translate tacit intentions into effective prompts, yielding brittle, inconsistent guidance, or slow down the promblem-sloving process (Zhang et al., 2025b). These limitations amplify variance across runs, introduce cognitive biases (e.g., anchoring on an early draft), and reduce reproducibility.

In contrast, FOR-Prompting encodes questioning, revision, and synthesis into agents with explicit, machine-executable objectives, enabling the system to interrogate and refine the intermediate solution without relying on ad hoc human expertise. This design is highly promising, as it not only reduces dependence on human intervention - often constrained by limited knowledge, incomplete problem understanding, or insufficient prompt-crafting skills - but also enhances the consistency, reproducibility, and transparency of the reasoning process. By transforming what was once an intuitive and manual collaborative process into a structured, automated workflow, FOR-Prompting offers a scalable and domain-agnostic framework for complex reasoning tasks. It preserves the strengths of human-style dialogue-driven refinement while overcoming the variability, cost, and cognitive bias introduced by manual prompting, ultimately paving the way toward more autonomous, self-improving reasoning systems.

### 6.2 FOR-PROMPTING IS HELPFUL AND SHOWS PROMISE FOR SMALL MODELS

On GSM8K we observe a recurring pattern. Single prompt baselines are brittle and often miss the correct answer. CoT improves correctness by eliciting intermediate steps. In our experiments FOR-Prompting reached accuracy comparable to CoT and produced reasoning that annotators viewed as more coherent and more responsive to targeted external questions. These lightweight questions act like a gentle adversary and push the solver to state assumptions and address salient gaps before finalizing the answer. This yields solutions that align better with human judgments.

Many real tasks do not have a single ground truth solution. In brainstorming, plan refinement, and strategy design, value comes from surfacing blind spots, stress testing feasibility, and moving from vague goals to concrete next steps. FOR-Prompting provides an automated scaffold for this kind of thinking. External questions guide exploration, push for specificity, and make trade offs explicit. The protocol therefore supports ideation that is more diverse, plans that are more actionable, and narratives that better document why a decision was made. As evaluation in these domains often relies

on human preference and rubric based judgments, the transparency of the dialogue trace becomes an asset rather than an overhead.

This mechanism may have substantial potential for small model applications (Wang et al., 2025). For a 1B parameter model the absolute accuracy on math problems is usually well below that of larger models, yet one brief round of external questioning typically brings a notable improvement over direct prompting. For planning and brainstorming tasks FOR-Prompting can also be attractive on smaller models. Our results indicate advantages in interaction quality, outcome quality, information coverage, and level of refinement. The remaining consideration is cost. The combination with small models is practical because it offers cost efficiency, on-device or constrained deployment, and privacy when local reasoning is required. Example scenarios include privacy sensitive assistants that must reason on-device, on-device personal use or research deployments with limited compute, and edge applications where intermittent connectivity or energy limits rule out large models.

The most immediate cost is token usage and latency. External questioning can increase both, yet the overhead is usually controllable. This overhead compares favorably to scaling parametric capacity. Users can iterate extensively on a personal device without incurring high API token costs.

An additional practical benefit emerges from our cross-model role-swapping experiments. When we assign the small model as the Debater (question generator) and use the stronger model only as the Defender, we observe that the overall reasoning quality remains high while the token usage of the larger model is greatly reduced. This asymmetric structure stands in contrast to self-ask–style approaches, where a single (often large) model must generate both sub-questions and answers, doubling the load on the expensive model. In our setting, the Debater only needs to generate short interrogative prompts, a task that even a 1B model can perform reliably. As a result, most of the computational cost is shifted away from the frontier model and toward the lightweight model, which is inexpensive to run locally. This makes FOR-Prompting attractive in scenarios where large-model usage must be minimized, such as cost-sensitive deployments, rate-limited API environments, or hybrid pipelines where local small models serve as question-generating front ends for stronger but more expensive solvers.

# 7 LIMITATIONS

Although FOR-Prompting offers improvements over some prompting strategies and reduces potential human labor costs in open-ended tasks, several limitations remain. For instance, the protocol introduces additional computational overhead. FOR-Prompting relies on multi-turn interactions between the Debater and the Defender, which increases latency compared with single-turn prompting methods such as standard prompting or CoT. As a result, the approach is less suitable for applications that require real-time responses or low-latency interactions. Instead, FOR-Prompting is better suited for tasks that can tolerate background processing and iterative refinement, such as open-ended reasoning, analysis, or planning, where response speed is not the primary constraint. Our study is limited by compute constraints, which prevented evaluation on more and larger open-source datasets. We were also unable to reproduce certain reported baselines for some models (e.g., LLaMA 3.2:1B), so all benchmark values are computed using our standardized pipeline.

# 8 CONCLUSION AND FUTURE WORK

FOR-Prompting targets the reasoning process by introducing structured external questioning that surfaces hidden assumptions and closes inferential gaps. This approach improves correctness and transparency at modest computational cost, enables the use of smaller models and on-device deployments, and remains backend-agnostic since it relies on role-based interactions rather than model-specific features. vspace-0.2em Future work will broaden the range of tasks and modalities, develop adaptive questioning policies, and integrate domain-specific validators to complement our accuracy and reasoning proxies. We also plan to release standardized prompts and transcripts to support reproducible benchmarking and evaluate FOR-Prompting in practical settings such as education, accessibility, and edge assistants. Additionally, future investigations may explore multi-agent variants of FOR-Prompting involving more than two coordinated roles, enabling richer forms of interrogative pressure and collaborative reasoning. We also aim to examine the generalizability of the protocol across languages, cultural contexts, and real-world open-ended decision-making scenarios. Another promising direction is integrating FOR-Prompting with retrieval-augmented generation (RAG) or reinforcement learning (RL). Combining external knowledge retrieval with structured interrogative pressure may strengthen long-horizon planning and multi-stage tasks.

## ETHICS STATEMENT

This work includes a minimal-risk human preference study. Participants were recruited voluntarily through social media and verified as adults via email-based confirmation for eligibility. After verification, all personally identifiable information was removed, and only anonymized preference responses were retained. All procedures follow commonly accepted ethical guidelines for minimal-risk user research.

## USE OF LLMs

We employed LLMs for limited purposes: (1) to improve the readability of this paper (grammar and spelling correction), and (2) to reformat outputs in the Appendix that could not be directly rendered in LaTeX. Some reported results (*coherence*, *reasoning*, and *GPT-4.1 evaluation*) were obtained using an LLM-as-a-judge approach, as noted in the main text, whereas accuracy scores were computed through statistical calculations.

## REPRODUCIBILITY STATEMENT

Because our contribution, FOR-Prompting, is primarily a framework-level design rather than a fixed implementation, the specific code and prompts used in our experiments do not need to be followed verbatim. Instead, we provide illustrative code snippets and representative outputs, which are sufficient to reproduce and verify the core results. These materials are made available through an anonymized repository (`https://anonymous.4open.science/r/as98f4-6cEE/`) during the review process and will be released publicly upon acceptance.

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

## A  APPENDIX

### A.1  EXAMPLE ROLE PROMPTS.

Table 4 shows example prompts for each role in the FOR-Prompting framework. It worth to note that the prompts listed for each role in the FOR-Prompting framework serve as illustrative examples. In practice, prompts with similar intent or wording can be employed to achieve comparable effects, and there is no requirement to use these exact phrasings.

| Role | Example Prompt |
|------|----------------|
| Defender | "You are the Defender. Answer the user's task as best as possible. Be concise but correct. Provide a single final answer at the end, clearly marked." |
| Debater | "You are the Debater. You will ask clarifying questions to improve the answer, without proposing any solutions. Your output should be ONLY a concise list of questions." |
| Host | "You are the Host. You must synthesize the Defender's answer and the Debater's objections, and produce a refined final answer. Quote specific objections when relevant." |

Table 4: Illustrative prompts for each role in FOR-Prompting.

### A.2  FULL OUTPUTS OF FOR-PROMPTING IN CASE STUDY 3 - TASK "HOW MANY RS IN STRARRTRABBBERY"

The following presents the full record of the task described in Case Study 3. We conducted seven rounds of iteration. Although the FOR-Prompting method produced the correct answer after the first round, later confirmed by human review, the subsequent iterations enhanced the interpretability of the answer, effectively providing six additional layers of verification from different perspectives (through questions). The final result synthesized by the "Host" was also correct.

**Task:**  How many rs in strarrtrabbbery

**Defender:**  "There are 4 'r's in 'strarrtrabbbery'."

**Debater:** "Are you sure there are only 4 'r's in 'strarrtrabbbery'? Counting manually, it appears there may be more than 4.", "Did you double-check the spelling of the word to ensure accuracy in your count?", "Could you show how you counted the 'r's to clarify your process?"

**Defender:** Let's count the 'r's in 'strarrtrabbbery' together, letter by letter:

s t r a r r t r a b b b e r y

Now, let's highlight the 'r's:

s t **r** a **r r** t **r** a b b b e **r** y

So, the positions of 'r' are: 3rd, 5th, 6th, 8th, and 13th letters.

Total number of 'r's: 5.

Therefore, there are 5 'r's in 'strarrtrabbbery'. Thank you for pointing out the need to double-check!

**Debater:** "Is the highlighting of 'r's consistent with the actual positions in the word 'strarrtrabb-bery'? For example, are the 5th and 6th letters both 'r'?", "Could you clarify why the 13th letter is marked as an 'r', and whether this matches the raw spelling?", "Would it be helpful to present the word with indices for better clarity in counting?", "Are there any possible typographical errors in the word or in the counting process that could affect the accuracy of the result?", "Is there a reason you thanked the user for double-checking, and could you provide more assurance by showing the count step by step with indices?"

**Defender:** Let's carefully examine the word 'strarrtrabbbery' step by step, addressing each critique:

**1. Listing with indices for clarity:**

| Index | Letter |
|-------|--------|
| 1 | s |
| 2 | t |
| 3 | r |
| 4 | a |
| 5 | r |
| 6 | r |
| 7 | t |
| 8 | r |
| 9 | a |
| 10 | b |
| 11 | b |
| 12 | b |
| 13 | e |
| 14 | r |
| 15 | y |

**2. Counting 'r's and their positions:**
– 3rd letter: r
– 5th letter: r
– 6th letter: r
– 8th letter: r
– 14th letter: r
So, the 'r's are at positions 3, 5, 6, 8, and 14.

**3. Are 5th and 6th letters both 'r'?**
Yes, as shown above, both the 5th and 6th letters are 'r'.

**4. Is the 13th letter an 'r'?**
No, the 13th letter is 'e', not 'r'. The 14th letter is 'r'.

**5. Typographical errors:**

If the spelling is exactly as you wrote—'strarrtrabbbery'—then the above count is correct. If there was a typo in the word, the count could change.

**6. Step-by-step count and assurance:**

By displaying the word with indices and listing all the 'r' positions, the count is accurate and transparent.

**Final count:** There are 5 'r's in 'strarrtrabbbery', at positions 3, 5, 6, 8, and 14.

**Debater:** "Does the listing with indices provide enough clarity, or could the formatting be improved for easier verification?", "Is there any ambiguity in how the indices correspond to the letters, particularly if the table or formatting is misaligned?", "Are the positions of the 'r's correctly identified, or is there room for error in transcribing the positions from the table to the list?", "Would it be beneficial to explicitly show the process of counting each 'r' in the word as it is written, rather than relying on the table?", "Is there an assumption that the word was spelled correctly, and how would the analysis change if there was a typographical error?", "Are there any additional checks that could be included to further ensure the transparency and accuracy of the count?", "Could the explanation include a visual or color-coded representation for even greater clarity?"

**Defender:** Let's improve the clarity and accuracy of counting the number of 'r's in "strarrtrabbbery" by addressing each critique:

**1. Improved formatting for easier verification:**

A simple inline format can help ensure indices match up with letters:

```
1  2  3  4  5  6  7  8  9 10 11 12 13 14 15
s  t  r  a  r  r  t  r  a  b  b  b  e  r  y
```

**2. Ambiguity in indices:**

Displaying both indices and letters in a single line as above prevents misalignment and makes it clear which index corresponds to each letter.

**3. Are 'r' positions identified correctly?**

Let's mark the 'r's directly:

```
 1   2    3    4    5    6   7    8    9   10   11   12   13   14   15
 s   t   [r]   a   [r]  [r]  t   [r]   a    b    b    b    e   [r]   y
          ↑         ↑    ↑        ↑                        ↑
```

The 'r's appear at positions: 3, 5, 6, 8, and 14.

**4. Explicitly showing the process:**

Walking through the word, letter by letter:

- 1: s (not r)
- 2: t (not r)
- 3: r (1st r)
- 4: a (not r)
- 5: r (2nd r)
- 6: r (3rd r)
- 7: t (not r)
- 8: r (4th r)
- 9: a (not r)
- 10: b (not r)
- 11: b (not r)

- 12: b (not r)
- 13: e (not r)
- 14: r (5th r)
- 15: y (not r)

Total 'r's: 5

**5. Assumption about spelling:**
This count assumes the word is spelled as given. If there is a typo, the count would change. Double-check the spelling before counting.

**6. Additional checks:**
Double-count by highlighting the 'r's, or use a tool (such as a find/count function) for large words.

**7. Visual or color-coded representation:**
If possible, color-code the 'r's for clarity (here, using brackets):

```
s   t   r   a   r   r   t   r   a   b   b   b   e   r   y
        ↑       ↑   ↑       ↑                       ↑
```

**Conclusion:**
There are 5 'r's in "strarrtrabbbery", at positions 3, 5, 6, 8, and 14. The improved formatting and explicit counting process make this transparent and easy to verify.

**Debater:** "Is the inline format with indices and letters actually make the process more transparent, or could it become confusing if not perfectly aligned (especially in plain text)?", "Is there still any risk of ambiguity in the indices if a proportional font or formatting error occurs?", "When marking the 'r's with brackets, is there a possibility that this notation could be misinterpreted or missed, especially for those unfamiliar with the method?", "Is it necessary to list every letter and its check, or would a more concise method suffice for such a short example?", "Does the answer adequately address what would happen if there were a typo, or should there be a suggested process for validating the original word's spelling before starting?", "Are there other efficient or automated ways to check the count, especially for longer words or strings?", "Would using color (if possible) or another visual aid significantly improve clarity, or does the current bracket system suffice for most scenarios?"

**Defender:** Certainly! Here's a refined response that directly addresses the critiques and provides both a clear answer and guidance for best practices:

**How many 'r's in 'strarrtrabbbery'?**

**Short, clear method:**
For a short word like "strarrtrabbbery", the simplest and least error-prone way is to scan the word and count each 'r':

```
s   t   r   a   r   r   t   r   a   b   b   b   e   r   y
        ↑       ↑   ↑       ↑                       ↑
```

Going through the word letter by letter:

```
s   t   r   a   r   r   t   r   a   b   b   b   e   r   y
        ↑       ↑   ↑       ↑                       ↑
```

– The 'r's appear at positions 3, 5, 6, 8, and 14 (counting from 1).
– **Total: 5 'r's.**

**Addressing the critiques:**

1. *Inline format with indices:* For very short examples, adding indices or brackets is helpful only if the spacing is exact (i.e., in monospaced font). In plain text or proportional fonts, alignment can be lost, making it confusing. For short examples, it's usually clearer just to highlight the letters or list their positions.

2. *Risk of ambiguity:* Yes, indices can become ambiguous if formatting is off. For longer or more complex strings, consider using tools or code to count.

3. *Brackets for marking:* Brackets can be missed or misunderstood if the reader isn't familiar with the method. Explicitly stating the positions or just listing the count is often clearer.

4. *Conciseness:* For a short string like this, listing every letter and its check is overkill. A simple count with a quick visual scan suffices.

5. *Typos and spelling:* Always check the spelling of the word before counting. If accuracy is critical, copy and paste the source or use a spell-checker.

6. *Automated/efficient methods:* For long strings, use a text editor's search feature (Ctrl+F) or a simple script (e.g., Python: `word.count(''r'')`).

7. *Visual/color cues:* If possible, use color or bold to highlight letters, but in plain text, a caret (ˆ) or simply stating positions is usually sufficient.

**Summary:**
– There are **5** 'r's in "strarrtrabbbery".
– For short words, scan and count. For long strings, use search or code.
– Always verify the spelling before you start.

**Debater:** "Is the caret marking method really effective outside of monospaced fonts, or could it introduce more confusion for most users?", "Does the response give enough guidance for users who find brackets or bolding visually cluttered—are alternative suggestions needed?", "When listing the positions of 'r's, is it clear enough whether counting starts from 1 or 0? Should an explicit example be given?", "Does the section on automated methods sufficiently warn about the risk of missing 'R' if the search is case-sensitive?", "For 'Scripts', should the answer mention what to do if the user doesn't know programming, or suggest user-friendly alternatives?", "Is it necessary to elaborate more on what to do if a typo is found—should the answer suggest correcting and recounting?", "Could the advice on counting in short words be more strongly phrased to highlight that errors are common even for short or simple words?", "Does the summary table help users choose the right method, or would an example scenario be more helpful?", "Should the response mention accessibility tools or features (like screen readers) that can help users with dyslexia or visual impairments count letters?", "Is the response's structure (with repeated examples and summary) redundant or does it aid comprehension? Would a more concise version be just as clear?"

**Defender:** Certainly! Here is an improved response that directly addresses all the critiques and clarifies approaches for counting 'r's in "strarrtrabbbery":

**How many 'r's in 'strarrtrabbbery'?**

**Short Answer:**
There are **5** occurrences of the letter 'r' in the word "strarrtrabbbery".

**Detailed Guidance:**

1. **Manual Counting: Reliable Basics**
   Read each letter one by one and tally every 'r' you encounter. You can use your finger, a pencil, or physically point to keep track. **Caution:** Even in short or simple words, repeated or clustered letters make mistakes surprisingly common. For best accuracy, count more than once or use a second method.

2. **Marking Methods: Pros, Cons, and Suggestions**
   *Carets (ˆ) above letters:*
   Example (works best in monospaced fonts only):
   ```
   strarrtrabbbery
   ˆ  ˆ ˆ ˆ       ˆ
   ```
   *Brackets [ ] or bolding:*
   s t [r] a [r] [r] t [r] a b b b e [r] y
   *Alternative:* underline/highlight or circle each 'r' on paper.

3. **Listing Positions: Be Explicit**
   The 'r's are at positions 3, 5, 6, 8, and 14 (counting from 1: s=1, t=2, r=3...). Always state whether you start at 1 or 0.

4. **Automated Methods and Case Sensitivity**
   Use Ctrl+F or editor search; check case sensitivity. For scripts, e.g., Python `word.count(''r'')` (beware hidden characters and case).

5. **If You Don't Code**
   Use an online letter counter; or ask someone to help with a quick script.

6. **Check for Typos Before Counting**
   Confirm spelling, correct if needed, then recount (copy-paste to avoid input errors).

7. **Accessibility Features**
   Consider screen readers, browser extensions for letter highlighting, or printing and marking physically if visual tracking is difficult.

**Summary Table (tabular format):**

| Method | When to Use | Cautions/Alternatives |
| --- | --- | --- |
| Manual count | Always possible | Double-check repeats/typos |
| Carets | Monospaced fonts | Avoid in variable-width fonts |
| Brackets/Bold | If not cluttered | Underline/highlight instead |
| Positions | Explicit reference | State start index (1/0) |
| Automated | Long/complex strings | Watch case sensitivity |
| Scripts | If you know how | Use online tools if not |
| Accessibility | If needed | Screen readers, print, aids |

**In summary:**
There are **5** 'r's in "strarrtrabbbery". Choose a method that fits your context and double-check if accuracy matters.

**Host:** Final Result
There are **5** 'r's in "strarrtrabbbery", at positions 3, 5, 6, 8, and 14 (counting from 1).

A.3 FULL OUTPUTS OF FOR-PROMPTING IN CASE STUDY 4 - TASK "CREATE A 5-DAY ITINERARY FOR A TRIP TO RIO DE JANEIRO IN APRIL."

**Task:** Create a 5-day itinerary for a trip to Rio de Janeiro in April.

**Defender:** Day 1: Arrival & Copacabana - Morning: Arrive in Rio de Janeiro, check into your hotel near Copacabana or Ipanema. - Afternoon: Relax and stroll along Copacabana Beach. Grab lunch at a beachside kiosk. - Evening: Walk to Forte de Copacabana for sunset views. Dinner at a local Brazilian steakhouse (churrascaria).

Day 2: Christ the Redeemer & Santa Teresa - Morning: Take the train up Corcovado Mountain to visit Christ the Redeemer. Enjoy panoramic city views. - Afternoon: Head to the bohemian Santa Teresa neighborhood. Explore its art studios, cafes, and colonial streets. - Evening: Have dinner in Santa Teresa and enjoy live samba music at a local bar.

Day 3: Sugarloaf Mountain & Lapa - Morning: Ride the cable car to Sugarloaf Mountain (Pão de Açúcar) for breathtaking views of Rio and Guanabara Bay. - Afternoon: Visit the Selarón Steps (Escadaria Selarón) and explore the vibrant Lapa district. - Evening: Experience Lapa's nightlife and try traditional Brazilian snacks and caipirinhas.

Day 4: Botanical Garden & Ipanema - Morning: Visit the Rio de Janeiro Botanical Garden (Jardim Botânico), famous for its lush flora and giant water lilies. - Afternoon: Relax at Ipanema Beach. Shop at the local boutiques or visit the Hippie Fair (if it's Sunday). - Evening: Dine at a seafood restaurant overlooking the ocean.

Day 5: Tijuca Forest & Departure - Morning: Take a guided hike in Tijuca National Park, the world's largest urban rainforest. See waterfalls and wildlife. - Afternoon: Last-minute shopping or a final stroll along the beach. - Evening: Depart Rio de Janeiro.

Tips: - Pack light clothes, sunscreen, and a hat for April's warm weather. - Stay hydrated and keep an eye on your belongings in crowded areas. - Try local dishes like feijoada, pão de queijo, and fresh tropical fruits.

**Debater:** "Day 1 seems quite relaxed, but is there enough time to check into the hotel and still enjoy Copacabana Beach and Forte de Copacabana if the arrival is late in the morning or early afternoon?",

"The recommendation to stay near Copacabana or Ipanema is good, but are there price or safety considerations travelers should be aware of for these neighborhoods?",

"Is Forte de Copacabana open late enough for sunset visits, and are there any entrance fees or restrictions that should be noted?",

"On Day 2, is it necessary to book tickets in advance for the Corcovado train to Christ the Redeemer, especially during busy travel seasons?",

"Santa Teresa is a great suggestion, but are there safety tips for visiting this area, and is public transport or a taxi recommended for getting there?",

"Live samba music in Santa Teresa is appealing, but are there specific venues or days when performances are more likely?",

"Day 3's plan includes Sugarloaf Mountain and Lapa; is it manageable to fit both in one day without feeling rushed, especially considering travel times and possible queues?",

"Lapa is famous for nightlife, but are there safety concerns or areas to avoid after dark, and are there recommended places for first-time visitors?",

"On Day 4, is the Hippie Fair only on Sundays, and if so, is there an alternative suggestion for non-Sunday visitors?",

"Visiting both the Botanical Garden and Ipanema Beach in one day may be ambitious; is there enough time to enjoy both without feeling hurried?",

"The seafood restaurant suggestion in Ipanema is nice; are there specific recommendations for popular or well-reviewed places?",

"For Day 5's hike in Tijuca National Park, is a guided tour essential for safety, or can experienced hikers go alone? Are there particular trails that are recommended for a half-day visit?",

"Are there recommendations on where to store luggage after hotel checkout if the flight departs in the evening?",

"The tips are practical, but could include advice on transportation (e.g., using taxis, ride-sharing, or public transit) and any relevant safety apps or emergency numbers.",

"Should travelers be aware of any major holidays, events, or closures in Rio during April that could affect this itinerary?",

"Are there any health or vaccination requirements for visiting Rio, especially for those planning outdoor activities in Tijuca Forest?",

"The itinerary includes many outdoor activities; is there a rainy season in April that could disrupt plans, and are there suggested indoor alternatives?",

"Are language barriers likely for English-speaking tourists, and are there tips for overcoming them (such as useful Portuguese phrases or translation apps)?",

"Does the itinerary take into account travel times between attractions, and would using a map app or planning routes in advance be helpful?",

"Overall, could the itinerary be adjusted for travelers with mobility issues or specific dietary restrictions?"

**Defender:** 5-Day Rio de Janeiro Itinerary (April) – Enhanced Version with Practicalities, Safety, and Alternatives

Day 1: Arrival, Copacabana Beach & Forte de Copacabana

- **Arrival:** If arriving late morning or early afternoon, plan for a quick hotel check-in (most hotels allow baggage drop before formal check-in time). Prioritize either Copacabana Beach or Forte de Copacabana if time is short; both are close together.

- **Copacabana/Ipanema Stay:** These are safe and lively neighborhoods but can be pricier. For budget travelers, check for hostels or guesthouses. Use hotel safes for valuables and avoid dark, deserted areas at night.

- **Forte de Copacabana:** Open until 6pm (last entry ˜5:30pm), entrance fee (about R$6). Sunset is often best viewed from the beach; for late arrivals, consider skipping the fort and enjoying the promenade.

Day 2: Christ the Redeemer & Santa Teresa

- **Corcovado Train:** Booking tickets in advance is highly recommended, especially on weekends or holidays (get them at tremdocorcovado.rio). Allow time for queues and transport; early morning is less crowded.

- **Santa Teresa:** Take a licensed taxi or app-based ride (Uber/99) for safety. Avoid walking alone after dark. Stay on main roads; keep valuables secure.

- **Live Samba:** Check venues like Bar do Mineiro (good food, sometimes live music), Armazém São Thiago, or Café do Alto. Fridays and Saturdays are most likely for performances; confirm with venues ahead.

Day 3: Sugarloaf Mountain & Lapa

- **Manageability:** Visit Sugarloaf in the morning (less crowded). Allow 2–3 hours, including queues. Lapa is a 20–30 min taxi ride from Sugarloaf. Manageable if you don't linger too long.

- **Lapa Safety:** Stick to main streets and avoid isolated alleys after dark. Recommended for first-timers: Rio Scenarium (iconic samba club), Lapa 40 Graus, or Selina Lapa Rooftop Bar.

Day 4: Jardim Botânico, Ipanema & Hippie Fair

- **Hippie Fair:** Only on Sundays. For other days, consider Feira de Copacabana (night street market) or boutique shopping on Rua Visconde de Pirajá.

- **Timing:** Botanical Garden (2–3 hours) + Ipanema Beach (rest of the day) is doable but don't rush. If tired, skip extra shopping.

- **Seafood:** Try Restaurante Satyricon, Zaza Bistro, or Barraca do Uruguai for well-reviewed seafood.

Day 5: Tijuca Forest Hike & Departure

- **Guided vs. Solo Hike:** Experienced hikers can do main trails solo, but for Pico da Tijuca or Pedra Bonita, a guide is safer (especially for first-timers). Shorter trails like Cascatinha Taunay are manageable alone.

- **Luggage Storage:** Most hotels offer luggage storage. Alternatively, use LuggageHero or airport lockers.

General Tips & Practicalities:

- **Transport:** Use Uber or 99 apps for safe, affordable rides. Metro is faster for some routes. Rio de Janeiro Metro app and Moovit are helpful. Emergency number: 190 (police). Download "Onde Tem Táxi" app for official taxis.

- **Weather:** April is autumn, warm (22–28°C), with some chance of rain. Have indoor alternatives: Museu do Amanhã (science museum), MAR (art museum), or AquaRio (aquarium).

- **Events/Holidays:** April can include Easter (Semana Santa); attractions may close or be crowded. Double-check opening hours in advance.

- **Health:** No mandatory vaccinations for Rio, but yellow fever vaccination is recommended for Tijuca Forest. Bring mosquito repellent.

- **Language:** English is limited. Download Google Translate; learn key phrases (e.g., "por favor," "obrigado/a"). Most tourist areas have English signage.

- **Mobility/Diet:** All attractions have some accessible options, but sidewalks may be uneven. Many restaurants cater to vegetarians/vegans; always ask about dietary needs.

- **Travel Times:** Use Google Maps or Moovit to plan routes and check for traffic. Allow extra time during rush hour.

This itinerary balances highlights with practical advice, safety tips, and alternatives for all travelers.

---

**Debater:** "Is Day 1 too ambitious for travelers who arrive later in the day, considering potential fatigue and limited daylight?", "While Copacabana and Ipanema are touted as safe, are there specific streets or zones within these neighborhoods that should be avoided, especially at night?", "Should the itinerary mention possible long lines or wait times at Forte de Copacabana, especially on weekends or holidays?", "On Day 2, are there alternative ways to reach Christ the Redeemer if Corcovado train tickets are sold out, such as vans or hikes?", "Does the Santa Teresa recommendation account for the steepness of the neighborhood, which may affect visitors with mobility issues?", "Are there reliable resources to check for updated samba schedules, since live music nights can vary and some venues may require reservations?", "Visiting Sugarloaf and Lapa on the same day assumes smooth transport—could traffic or public transit delays make this plan too rushed?", "Is it safe to visit Lapa during daytime, or is it best reserved for evenings and nightlife?", "For Day 4, should more detail be given on how to reach the Hippie Fair or alternative markets by public transport?", "Does the schedule for Day 4 allow enough downtime for relaxation, or does it risk overpacking the day with activities?", "The seafood recommendations are helpful, but are these restaurants accessible to non-Portuguese speakers and do they offer menus in English?", "Regarding the Tijuca Forest hike, are there entrance fees, opening hours, or permit requirements travelers should be aware of?", "Is luggage storage at hotels always free, or are there fees travelers should budget for when using services like LuggageHero or airport lockers?", "Should the transport section mention the risk of traffic jams at rush hour, especially when traveling to or from the airport?", "Are there indoor alternatives suggested for all areas of the city in case of rain, or are some neighborhoods lacking in rainy-day options?", "Does the itinerary consider public holidays or local festivals that might disrupt the plan beyond Easter?", "Are any health precautions missing, such as the risk of foodborne illness or sun exposure?", "How easy is it to find vegetarian or vegan food in traditional Brazilian restaurants, and are there specific places that cater to these diets?", "For travelers with limited mobility, are the main attractions truly accessible, or do some have stairs/steep paths that could pose problems?", "Does the itinerary suggest any tools for offline navigation, in case cell service is spotty or travelers lack data access?"

---

**Defender:** 5-Day Rio de Janeiro Itinerary in April (Refined with Critique Responses)

Day 1: Arrival & Copacabana/Ipanema - If you arrive late: Focus on a relaxed beach stroll (Copacabana Promenade or Ipanema's Posto 9), sunset, and a light meal. Forte de Copacabana closes at 6pm and often has a wait on weekends/holidays—skip if arrival is after 3pm or you're tired. - Safety: Copacabana and Ipanema are generally safe on the main avenues (Avenida Atlântica, Rua Visconde de Pirajá). Avoid the inner blocks near favelas (e.g., parts around Cantagalo, Pavão-Pavãozinho, and Arpoador at night). Stick to areas with good lighting and crowds after dark.

Day 2: Christ the Redeemer & Santa Teresa - Christ the Redeemer: If train tickets are sold out, official vans leave from Praça do Lido (Copacabana), Largo do Machado, and Barra da Tijuca. There's also a well-marked hiking trail (Parque Lage), but it's steep and best for fit travelers. - Santa Teresa: The area is charming but steep and cobbled—difficult for those with mobility issues. The Santa Teresa tram (bondinho) is an accessible, scenic way to reach the area from downtown. Taxis or ride-hailing apps are recommended for easier access. - Samba: For live music, check

sites like Sympla (event tickets), RioTur's agenda, or venues' social media for up-to-date schedules. Rio Scenarium and Bar Carioca da Gema are popular but may require reservations, especially on weekends.

Day 3: Sugarloaf Mountain & Lapa - Timing: Visit Sugarloaf early to avoid lines (opens 8am). Allow 2–3 hours. Lapa is best in the evening for nightlife, but the famous arches and Escadaria Selarón are safe to visit by day. Allow for 30–45 minutes commute between the two—consider traffic at rush hour (4–7pm). If running late, prioritize one. - Safety: Lapa is safe during the day on main streets (Avenida Mem de Sá, Rua do Lavradio) but avoid wandering late at night unless in groups or with a guide.

Day 4: Jardim Botânico, Ipanema & Hippie Fair - Hippie Fair: Only on Sundays. For other days, consider Feira de Copacabana (night street market) or boutique shopping on Rua Visconde de Pirajá. - Seafood: Satyricon, Zaza Bistro, and Barraca do Uruguai are popular with English menus and staff used to tourists. Reservations suggested on weekends. - Downtime: The afternoon is reserved for beach relaxation or hotel pool time. Activities are optional and can be dropped if you need more rest.

Day 5: Tijuca Forest Hike & Departure - Hiking: Parque Nacional da Tijuca is free, open daily 8am–5pm. No permits needed for main trails. Guided tours are best for Pico da Tijuca or Pedra Bonita (available via GetYourGuide, Viator, or local agencies). Shorter trails like Cascatinha Taunay are accessible and scenic. - Luggage: Most hotels offer free luggage storage, but confirm in advance. LuggageHero and airport lockers charge by the hour/day (typically R$30–60 per bag).

General Tips - Transport: Metro and ride-hailing apps are safest. Allow extra time during rush hour, especially to/from the airport. - Rain: For bad weather, consider Museu do Amanhã (downtown), MAR (art museum), Shopping Leblon, AquaRio (aquarium), or CCBB (cultural center) in Centro. Each is easily reached by metro or taxi. - Public holidays: Beyond Easter, Tiradentes Day (April 21) may affect opening hours. Confirm with venues in advance. - Health: Use bottled water, avoid undercooked street food, and use sunscreen (UV is high even in April). Carry insect repellent for hikes. - Dietary needs: Vegetarian/vegan options are limited at traditional restaurants, but spots like Teva, Prana, and Org Bistro cater to these diets. Most places in tourist areas offer English menus or at least English-speaking staff. - Accessibility: Christ the Redeemer, Sugarloaf, and Jardim Botânico have accessible paths and elevators, but Santa Teresa and the Selarón steps are challenging. Always check with venues if you have mobility concerns. - Offline navigation: Download Google Maps for Rio, or use Maps.me. Rio Metro app works offline for metro planning.

---

**Debater:** "Does Day 1 provide enough flexibility for travelers arriving very late in the day, or should it suggest skipping activities entirely if arrival is after sunset?", "The safety advice is helpful, but could it be more specific regarding exactly which streets or blocks are best avoided, especially for those unfamiliar with the area?", "On Day 2, while van alternatives for Christ the Redeemer are mentioned, is there information about how to purchase tickets for these vans and potential wait times?", "Santa Teresa is noted as steep and difficult for those with mobility issues, but does the itinerary provide enough alternatives or modifications for travelers with disabilities?", "Are the samba venue recommendations up to date, and is there a risk of venues being closed or having limited schedules outside of peak tourist times?", "For Day 3, is the suggested timing between Sugarloaf and Lapa realistic, considering possible delays or the need for rest between activities?", "Does the itinerary sufficiently warn about petty crime or pickpocketing in tourist-heavy areas like Lapa and Sugarloaf?", "On Day 4, is the Hippie Fair included only on Sundays? Are there enough alternative options for markets on other days, or does the plan become thin?", "Are the seafood restaurant recommendations suitable for travelers with dietary restrictions or allergies, and is there a price range provided?", "Does the itinerary on Day 4 risk being too crowded with activities, or is there enough downtime for relaxation?", "For Day 5, is it clear how to access guided tours for the hikes, and are there options for non-Portuguese speakers?", "Does the luggage advice account for possible hotel policies that charge for storage, or should travelers always expect to pay unless confirmed otherwise?", "Are the rain-day activity suggestions spread out geographically, or would visitors have to travel far in poor weather to reach them?", "Should the plan address how to handle sudden closures or schedule changes due to public holidays more proactively?", "Are health safety tips adequate, or should there be more detail about local illnesses, water safety, or emergency procedures?", "While vegetarian/vegan options are named, are there enough suggestions for those with other dietary needs

(e.g., gluten-free, allergies)?", "Is the summary of accessibility for major attractions accurate, and should there be more detailed guidance for wheelchair users or those with significant mobility issues?", "Does the itinerary provide enough detail about how to use offline navigation tools, such as step-by-step instructions for downloading maps?", "Are there any potential gaps in transport advice, such as how to safely use taxis or ride-hailing apps and what to do if public transit is disrupted?", "Should the itinerary include advice about local customs, tipping, or language barriers to help travelers better prepare?"

---

**Defender:**   5-Day Rio de Janeiro Itinerary in April (Refined for Clarity, Detail, and Practicality)

### Day 1: Arrival, Copacabana & Ipanema

- Late Arrivals: If your flight arrives after 7pm, skip all sightseeing and beach activities entirely for safety and rest. Use this time only for checking in, settling, and eating at your hotel or a nearby, well-lit restaurant. If you arrive even later (e.g., after 10pm), go straight to your accommodation and rest—do not attempt any itinerary activities.

- Safety: For Copacabana, limit yourself to Avenida Atlântica and avoid backstreets behind the hotels after sunset. In Ipanema, stay on Rua Visconde de Pirajá and along the main beach promenade; avoid walking towards the hills (especially near Cantagalo or Pavão-Pavãozinho) at night. Use app-based transportation at night and do not walk alone on the sand after dark.

### Day 2: Christ the Redeemer & Santa Teresa

- Christ the Redeemer: Purchase van tickets only from official counters at Praça do Lido, Largo do Machado, or Barra Shopping, or online at `https://paineirascorcovado.com.br` (choose "oficial van"). Arrive before 8am for the shortest lines; expect 30–90 minutes' wait if later. The official van is the only step-free option (the train is not wheelchair accessible). Bring ID for ticket collection. There are bathrooms and snack bars at the base and summit.

- Santa Teresa Mobility: Take the historic tram (bonde) from Centro (wheelchair accessible). For those with limited mobility, focus your visit on Largo dos Guimarães, which is relatively flat and offers cafes and art shops. Skip the Selarón Steps (steep) if needed. For fully accessible alternatives, consider the Museum of Tomorrow or the Rio Art Museum (both fully accessible and with step-free access).

- Samba Venues: To verify hours, check the venues' official websites and Instagram pages. For proactive planning, use the Rio nightlife aggregator "Sympla" or call ahead. Have backup options such as live music bars in Lapa or dinner at a traditional restaurant if samba venues are closed.

### Day 3: Sugarloaf Mountain & Lapa

- Timing: Plan 2–2.5 hours for Sugarloaf (including cable cars and views), then a minimum 45 minutes for transit to Lapa. Schedule a relaxed lunch in between to prevent feeling rushed. If tired, skip Lapa or move it to the evening of another day.

- Crime Risks: At Sugarloaf, keep valuables zipped and never set belongings down. In Lapa, avoid side streets and dark areas after dusk; stick to Rua do Lavradio and Avenida Mem de Sá. Use taxis/rideshare after dark. Pickpocketing is the main risk—keep phones and wallets secure and be cautious if approached by strangers.

### Day 4: Jardim Botânico, Hippie Fair & Markets

- Market Alternatives: If the Hippie Fair isn't running, try the Feira de São Cristóvão (Northeastern market), Feira da Glória (Sunday), or the antique market at Praça XV. Or, explore artisanal shops in Leblon or the Ipanema Farmers' Market (Fridays). For more variety, consider a food tour or chocolate tasting class.

- Dietary Needs: Most restaurants can accommodate allergies if given advance notice. Use allergy cards (in Portuguese), and ask for the chef if you have multiple allergies. Zazá Bistro, Da Bela, and Org Bistrô are known for handling complex dietary needs. Always double-check if there is any doubt, and bring snacks if you have severe allergies.

- Downtime: To avoid overscheduling, schedule pool or beach time, a spa massage (many hotels offer), or a leisurely cafe visit. Consider a sunset at Arpoador or a relaxed boardwalk stroll. Remind yourself that breaks are part of the Rio experience!

**Day 5: Accessible Options for Non-Hikers**

- In addition to Parque Lage and rooftop brunches, consider the Museu do Amanhã (fully accessible), a guided street art tour by car, or a cooking class in Ipanema. For families, the AquaRio aquarium is also accessible and engaging.

- Luggage Storage: Most hotels provide free storage, but always confirm at check-in. For third-party services like Bounce or LuggageHero, read user reviews and check for insurance. Have a backup plan: shopping mall lockers (e.g., Rio Sul), or ask your final excursion provider if they can hold bags. If all else fails, the airport has reliable paid lockers.

**General Guidance**

- **Rainy Day Options:** Besides museums and malls, try niche indoor activities—Samba dance class, pottery workshop, escape room, or cinema with English subtitles (Kinoplex Leblon or Estação Net).

- **Holiday Closures:** Use Google Maps and check Instagram for real-time updates. For more proactive planning, download the "Guia Rio" app or use "Feriados RJ" for public holiday info. Ask your hotel to confirm openings the night prior.

- **Health & Safety:** In addition to dengue and food sensitivities, caution about sun exposure—use SPF 50+, hats, and hydrate constantly. Tap water is not potable; drink bottled/filtered only. Pharmacies ("Farmácia") are everywhere for supplies. Drink bottled or filtered water; avoid street food if you have a sensitive stomach.

- **Accessibility:** For tailored advice, contact Accessible Rio or Turismo Acessível. Major sights (Sugarloaf, Christ, Jardim Botânico) are accessible; Santa Teresa and Selarón Steps are not. Most Ubers can accommodate folding wheelchairs; for more, ask your hotel.

- **Offline Navigation:** Download Google Maps for offline use. Older travelers should request printed maps from hotels or tourist offices, or ask staff to print directions for each day's route.

- **Transport:** At airports/bus stations, if English support is lacking, look for official taxi stands (with price lists), or use Uber (set pick-up indoors). If in doubt, approach the information desk. If stuck, show your hotel address to staff and ask for help in simple English or via Google Translate.

- **Flexibility:** If weather or closures disrupt plans, swap beach/park days with museum/market days. Keep a list of rainy-day options. Don't hesitate to ask your hotel for daily local advice.

- **Balance of Experiences:** This itinerary now mixes major sights with authentic experiences (food tours, local markets, Santa Teresa, art walks) and downtime. Adjust daily based on your interests and energy—Rio rewards both adventure and relaxation.

---

**Host:  5-Day Rio de Janeiro Itinerary for April (Final Version)**

—

**Day 1: Arrival, Copacabana & Ipanema**

- **Arrival:** If arriving late, prioritize a relaxed stroll along the Copacabana or Ipanema beachfront promenade. Enjoy a light meal at a well-reviewed, nearby restaurant. Avoid poorly lit or unfamiliar areas after dark, and consult your hotel for safe walking routes.

- **Hotel Check-in:** Most hotels allow baggage drop before formal check-in. Use hotel safes for valuables.

- **Evening:** If time allows, catch the sunset from the promenade or Arpoador Rock (between Ipanema and Copacabana). For dinner, try a local churrascaria (Brazilian steakhouse) or a seafood restaurant.

- **Safety:** Main avenues are generally safe, but be vigilant for pickpocketing and avoid favela-adjacent areas at night. Ask your hotel for a safety map or advice.

—

**Day 2: Christ the Redeemer & Santa Teresa**

- **Morning:** Visit Christ the Redeemer. Book tickets for the Corcovado train in advance (`https://www.tremdocorcovado.rio/en/`). If sold out, use official vans from Praça do Lido (Copacabana), Largo do Machado, or Barra da Tijuca. Guided tours are also available via hotels.

- **Accessibility:** The train and vans are accessible. The hiking trail from Parque Lage is steep and best for fit travelers.

- **Afternoon:** Explore Santa Teresa's artistic neighborhood. Use the historic tram (bondinho) from downtown for a scenic, accessible ride. Visit Largo dos Guimarães, Parque das Ruínas (mostly accessible), and local art studios.

- **Evening:** Dine in Santa Teresa (Bar do Mineiro, Café do Alto). For live samba, check Rio Scenarium or Bar Carioca da Gema (reserve ahead, especially on weekends).

- **Safety:** Use taxis or ride-hailing apps for transport, especially after dark. Stick to main streets and avoid isolated areas.

—

**Day 3: Sugarloaf Mountain & Lapa**

- **Morning:** Ride the cable car up Sugarloaf Mountain (Pão de Açúcar). Arrive early (opens 8am) to avoid crowds. Allow 2–2.5 hours for the visit.

- **Afternoon:** Head to Lapa. Visit the Selarón Steps (bottom entrance on Rua Joaquim Silva is more accessible) and admire the Lapa Arches.

- **Evening:** Experience Lapa's nightlife—try Rio Scenarium, Lapa 40 Graus, or Selina Lapa Rooftop Bar. Sample caipirinhas and traditional snacks.

- **Safety:** Stick to main streets (Avenida Mem de Sá, Rua do Lavradio). Be wary of scams and keep valuables concealed. For emergencies, dial 190 (police).

—

**Day 4: Jardim Botânico & Markets** - **Morning:** Visit the Rio de Janeiro Botanical Garden (Jardim Botânico). Allow 2–3 hours to explore the lush flora and giant water lilies.

- **Afternoon:** Relax at Ipanema Beach. If it's Sunday, visit the Hippie Fair (Praça General Osório). On other days, try Feira de Copacabana (night market), Feira de São Cristóvão (Northeastern market), or artisanal shops in Leblon.

- **Evening:** Dine at a seafood restaurant (Satyricon, Zazá Bistro, Barraca do Uruguai). Many restaurants accommodate dietary needs—use allergy cards in Portuguese and notify staff in advance.

- **Downtime:** Schedule time for a pool, spa, or leisurely cafe visit. Sunset at Arpoador is a local favorite.

—

**Day 5: Tijuca Forest or Accessible Alternatives & Departure**

- **Morning:** For hikers, take a guided hike in Tijuca National Park (Pico da Tijuca or Pedra Bonita). For a lighter option, stroll in Parque Lage or visit the Museu do Amanhã (fully accessible).

- **Non-hikers:** Consider a rooftop brunch, street art tour by car, cooking class, or AquaRio aquarium (accessible and family-friendly).

- **Afternoon:** Last-minute shopping, a final beach stroll, or relaxation at your hotel.

- **Luggage Storage:** Most hotels offer storage. For third-party options, use Bounce, LuggageHero, or mall lockers (e.g., Rio Sul). Confirm details at check-in.

- **Evening:** Depart Rio de Janeiro.

—

**General Tips & Practicalities**

- **Transport:** Use Uber/99 for safe, affordable rides. Metro is efficient for some routes; download the Rio Metro or Moovit app. At airports/bus stations, use official taxi stands or ride-hailing apps—never accept rides from solicitors.

- **Weather:** April is warm (22–28°C) with some rain. Pack light clothes, sunscreen (SPF 50+), a hat, and mosquito repellent. Have indoor alternatives for rainy days: Museu do Amanhã, MAR, AquaRio, samba class, or a food tour.

- **Safety:** Stay hydrated, drink only bottled/filtered water, and keep an eye on belongings in crowds. For emergencies, dial 190 (police), 192 (ambulance). Pharmacies ("Farmácia") are widely available.

- **Language:** English is limited; download Google Translate and learn basic Portuguese phrases. Most tourist sites have some English signage.

- **Accessibility:** Major attractions (Sugarloaf, Christ, Jardim Botânico, Museu do Amanhã) are accessible. Santa Teresa and Selarón Steps are not. For tailored advice, contact Accessible Rio or Turismo Acessível.

- **Offline Navigation:** Download Google Maps for offline use. Request printed maps or directions from your hotel if needed.

- **Flexibility:** Swap beach/park days with museums/markets if weather or closures disrupt plans. Keep a list of rainy-day options and ask your hotel for daily local advice.

- **Dietary Needs:** Most restaurants can accommodate allergies and dietary restrictions. Always double-check with staff and carry snacks if you have severe allergies.

- **Events/Holidays:** April may include Easter (Semana Santa); check attraction hours in advance. Use Google Maps, Instagram, or the "Guia Rio" app for real-time updates.

—

**This itinerary balances Rio's iconic sights, authentic local experiences, downtime, and practical safety and accessibility advice—ensuring a memorable and enjoyable visit for all travelers.**

---

A.4 FULL OUTPUTS OF FOR-PROMPTING IN CASE STUDY 5 - THE EXTENSION OF CASE STUDY 4 WITH AN EXTRA CONDITION – "ON THE DAY-4, OTHER TRAVELERS WILL JOIN US, FORMING A GROUP OF SIX PEOPLE, AND WE WILL DO ACTIVITIES TOGETHER. PLEASE UPDATE THIS ITINERARY ACCORDINGLY."

**Task:** On the Day-4, other travelers will join us, forming a group of six people, and we will do activities together. Please update this itinerary accordingly.

---

**Defender:** Day 4: Jardim Botânico & Markets (Group Day)

- Group Logistics: With six people, plan for group-friendly transportation. Consider booking a large Uber, two taxis, or a private van for the day (many hotels can arrange this). Agree on a morning meeting time/location, and confirm everyone's dietary needs and interests in advance.

- Jardim Botânico: Arrive early (opens at 8 AM) to avoid crowds and maximize cooler morning hours. Group discounts are available if you purchase tickets together at the entrance. For a more engaging visit, book an English-speaking guided tour (ask at the visitor center or pre-arrange online). The gardens are mostly accessible with wide paths, but let the group know there is some walking involved. Bring water, hats, and sunscreen for everyone.

- Markets & Food: After the gardens, head as a group to a nearby market. If it's Sunday, the Feira Hippie de Ipanema is ideal for groups—plenty of space, food stalls, and shopping. For weekdays, consider the Feira de São Cristóvão (huge, lively, covered, and perfect for sampling Northeastern Brazilian food together). For smaller, more upscale experiences, visit artisanal shops in Leblon or the Ipanema Farmers' Market (Friday mornings). If the group is interested in food, book a guided food tour or a chocolate tasting workshop—these experiences are fun in groups and can be reserved in English.

- Lunch: Reserve a table for six in advance, especially on weekends. Options include Zazá Bistro (Ipanema, great for dietary restrictions), Da Bela (Botanical Garden area, farm-to-table), or a classic churrascaria (Brazilian steakhouse) like Fogo de Chão (Botafogo) for a festive group meal. Call ahead to confirm space and allergy accommodations.

- Downtime & Social Time: After lunch, plan some downtime together—pool time at your hotel, a relaxed walk on the Ipanema or Copacabana boardwalk, or a group spa session (some hotels offer group massage packages). For sunset, Arpoador is a popular, safe spot for groups—arrive early to find space.

- Evening (Optional): If energy allows, consider a group samba class (many studios offer private lessons for groups), or a casual bar/rooftop in Ipanema or Leblon for drinks and city views. Reserve if possible for a group this size.

- Group Safety & Coordination: - Stay together, especially in crowded markets. - Appoint a group leader or WhatsApp group for communication. - Review emergency meeting points. - Carry some cash (small bills) for markets and tips. - If splitting up, agree on a regrouping time/location.

Summary of Adjustments: - All activities are now group-friendly—transport, tours, meals, and downtime are optimized for six people. - Emphasis on advance reservations for tours and restaurants. - Market and food options tailored for group tastes and dietary needs. - Safety and communication tips for group travel. - Built-in flexibility for the group to split or relax together as needed.

Let me know if you want suggestions for a specific group activity (e.g., cooking class, private boat ride, samba night) or further restaurant recommendations for groups!

---

**Debater** "For group logistics, did you consider the potential for traffic delays or the availability of large Ubers or private vans during peak times? Are there backup options if these aren't available?", "While confirming dietary needs and interests in advance is wise, did you provide a clear method for gathering this information from the group?", "For Jardim Botânico, did you check if English-speaking guides are always available on the spot, or should booking in advance be mandatory during high season?", "You mention group discounts at the gardens—do you have details on what constitutes a group, or if prior arrangement is needed for the discount?", "When suggesting to bring water, hats, and sunscreen, do you have advice on where to purchase these locally if someone forgets?", "For market visits, did you clarify the distance and transportation options between Jardim Botânico and each market? Some markets (like Feira de São Cristóvão) are far from the gardens.", "Are there specific recommendations for group-friendly food stalls at these markets, or advice on seating availability for larger groups?", "The artisanal shops and farmers' market suggestions are good, but are these truly feasible for a group of six, or might they be too cramped or limited in offerings?", "About booking food tours or chocolate workshops, can you specify reputable providers or whether these require significant advance notice?", "For lunch, do you have recommendations in case the suggested restaurants are fully booked or not available for groups? For churrascarias, did you mention the price range or possible long wait times?", "Regarding downtime, not all hotels have pools or spa packages—do you have alternatives for groups staying in budget accommodations?", "For the sunset at Arpoador, are there safety concerns after dark, or tips on avoiding crowds, especially for first-time visitors?", "In the evening, are samba classes for groups commonly offered in English, and do they require advance reservations, or can they be arranged on short notice?", "Your safety advice is solid, but is there a recommendation for how to handle valuables or what to do if someone gets lost?", "The summary mentions flexibility to split up, but are there suggested ways to ensure everyone stays engaged and doesn't feel left out if interests diverge?", "Overall, while the plan is comprehensive, is there any advice for balancing structured activities with free time, or for dealing with group decision fatigue?"

---

**Defender** Here are updated, practical answers to your group-focused questions for Day 4 and the overall itinerary:

1. Group Logistics & Transport: Yes, traffic delays are common, especially midday and late afternoon. For six people, large Ubers (UberXL), two standard Ubers/taxis, or a pre-booked van are your main options. For peak times, book transport in advance via your hotel or apps like 99 Taxi. As a

backup, use radio taxis (ask hotel front desk), or public transit if comfortable—Botanic Garden and Ipanema are on main bus lines. Always allow 20–30 extra minutes for traffic.

2. Gathering Dietary Needs/Interests: Before the trip or each day, send a group WhatsApp/Signal message or quick Google Form (simple tick-boxes: allergies, preferences, must-sees, energy level for the day). At breakfast, do a verbal check-in to confirm.

3. Jardim Botânico Guides: English-speaking guides are sometimes available on the spot, but in high season (April is busy), book in advance via the official website or email (visit their site for contacts). Hotels can also arrange this.

4. Group Discounts at Jardim Botânico: As of 2024, groups of 5+ may qualify for a discount, but prior arrangement is often required (book online or call ahead). Bring ID for all participants. Confirm discount conditions on the official website as policies can change.

5. Buying Water, Hats, Sunscreen: Forgot something? Buy these at the Botanic Garden entrance kiosks, nearby pharmacies (Drogaria Pacheco, Raia), or supermarkets like Zona Sul (found in Ipanema, Jardim Botânico, Leblon). Prices are reasonable, and staff usually know basic English.

6. Market Distances & Transport: Ipanema Hippie Fair is a 15-min drive from the Botanic Garden; Feira de São Cristóvão is 30–45 min by car (longer in traffic). For group comfort, use UberXL or a van. Public buses run between the Garden and Ipanema/Leblon, but not directly to São Cristóvão (a taxi or Uber is best).

7. Group-Friendly Food Stalls/Seating: At Hippie Fair and São Cristóvão, look for larger food tents or open-air restaurants—many have communal tables and accept groups. For São Cristóvão, try "Barraca da Chiquita" (traditional food, group tables). At Hippie Fair, "Adega do Pimenta" and "Bar do Alemão" are group-friendly. Arrive early for lunch to secure seating.

8. Feasibility of Artisanal Shops/Farmers' Markets: Artisanal shops in Leblon/Ipanema are small but doable if you split up in pairs or browse at staggered times. Farmers' markets are bustling but not cramped—groups can navigate if they stay flexible and don't expect to sit. For a sit-down, prioritize markets with food courts (see above).

9. Booking Food Tours/Chocolate Workshops: Reputable providers include Eat Rio (food tours), Rio4Fun (culinary experiences), and Choco.Rio (chocolate workshops). Book at least 48–72 hours in advance for a group of six. Confirm English-speaking guides.

10. Lunch Backup Plans: Always have a Plan B and C. If Zazá Bistro, Da Bela, or a churrascaria is booked out, try Delírio Tropical (casual, healthy, many locations), Garota de Ipanema (classic, group tables), or Galeto Sat's (local rotisserie chicken). Churrascaria prices: R80–R150/person, not including drinks; go early or reserve to avoid long waits.

11. Downtime Alternatives for Budget Stays: If no hotel pool/spa, go to a public beach (Ipanema, Leblon), relax at Parque Lage's café (beautiful garden setting), or find a local juice bar (Bibi Sucos, Polis Sucos) for a group hangout. Public parks are safe during daylight.

12. Arpoador Sunset Safety & Crowds: Go before 5 PM to find a spot; leave by 7 PM. It's safe in groups, but watch for pickpockets and avoid lingering after dark. Stick to the main path, avoid side alleys, and keep small bags zipped.

13. Samba Classes for Groups: Many studios (e.g., Rio Samba Dancer, Casa de Dança Carlinhos de Jesus) offer group classes in English—book at least a day ahead, especially for private sessions. Some host drop-in classes, but reservations are safer for six people.

14. Handling Valuables & Lost Persons: Each person carries only what they need. Use crossbody bags, keep phones away in crowds, and leave passports at the hotel safe. Share a group location via WhatsApp. If someone gets lost, agree on a meeting point (e.g., market entrance) or call the hotel for assistance.

15. Engagement & Splitting Up: Use WhatsApp group chat for updates. If interests diverge, split into pairs/trios for an hour, then regroup for meals or main activities. Plan set meeting times/places, and encourage everyone to share their must-see picks ahead of time so no one feels left out.

16. Balancing Structure & Free Time: Start with a flexible plan and build in 1–2 hours of "free roaming" or rest. Rotate who leads the day or chooses lunch. Review the next day's plan together each evening to avoid decision fatigue and keep energy positive.

Let me know if you need more details or sample WhatsApp group check-in messages!

---

**Debater:** "For group transport, do you have a contingency plan if UberXLs or vans are unavailable at peak times, or if there's a shortage due to high demand?", "When suggesting dietary needs collection, do you have an example or template for the Google Form or WhatsApp message to streamline this process?", "Regarding Jardim Botânico guides, what is the process if an English-speaking guide is not available on the spot, and can the hotel guarantee booking one in advance?", "For group discounts at Jardim Botânico, can you clarify what documentation is needed and how far in advance arrangements must be made to secure the discount?", "While you mention where to buy essentials like water and sunscreen, are these locations reliably open early in the morning or late in the afternoon when visitors might arrive or leave?", "Is traffic between the Botanic Garden and markets like Feira de São Cristóvão a significant concern at specific times, and should groups budget more time for these transfers?", "You mention communal tables at markets, but are reservations possible at these venues, or is seating strictly first-come, first-served?", "For artisanal shops and farmers' markets, will a group of six moving together disrupt the flow or experience, and are there recommendations for how to split up effectively?", "Are the food tour and chocolate workshop providers verified for safety and quality, and do they cater specifically to English-speaking groups, especially during busy periods?", "Regarding lunch plans, do your backup restaurants require reservations for groups, and are they located close to the main attractions to minimize travel time?", "For downtime, are public beaches and parks safe for tourists in the late afternoon or early evening, and are there any security tips beyond general caution?", "At Arpoador, are there any facilities like public toilets or refreshment stands, and is there a risk of overcrowding that might make the sunset experience less enjoyable?", "Do samba class studios offer flexible scheduling for tourists, or do group classes require strict advance booking, especially for English-speaking instruction?", "In the event someone gets lost, is there a recommended protocol beyond meeting points and WhatsApp, such as contacting local authorities or tourist assistance?", "When splitting up, how can the group ensure everyone is included in decisions and no one feels isolated, especially if interests or energy levels differ widely?", "For balancing structure and free time, what are some strategies to handle disagreements about pacing or activity choices to keep group morale high?"

---

**Defender:** Here are detailed answers to your group travel questions for Day 4 (and relevant overall tips):

1. Contingency Plan for Group Transport at Peak Times: - If UberXLs/vans are unavailable, have a list of reliable radio taxi companies (e.g., Coopatur, Coopertáxi, Taxi Leblon). Ask your hotel for the numbers or to call on your behalf. - Consider splitting into two regular Ubers/taxis if needed (pre-agree on destination). - For longer trips (like to Feira de São Cristóvão), ask the hotel to book a private van or car service a day ahead. - Public transit is an option (bus lines 409, 439 from Botanic Garden to Ipanema/Leblon), but for six people, taxis/Ubers are faster and safer.

2. Dietary Needs Collection Template: - WhatsApp Example: "Hi everyone! Please reply with any food allergies, dietary restrictions, or strong dislikes (e.g., nuts, gluten, vegetarian, etc.). Also, let us know any food you're excited to try!" - Google Form Example Fields: Name / Allergy (Y/N, list if yes) / Dietary Preference (Vegan, Vegetarian, No Pork, etc.) / Foods to Avoid / Foods to Try / Emergency Contact.

3. Jardim Botânico English Guide Process: - Hotel can request/book an English-speaking guide 2–5 days in advance via the Jardim Botânico website or by phone. Same-day walk-in is risky for English guides, especially in busy season. - If not available, ask the hotel to arrange a freelance guide, or use an app like ToursByLocals or GetYourGuide.

4. Documentation for Group Discount: - Typically, you'll need a list of names and ID numbers. Some sites may require passports (copies are fine). Book at least 3–5 days in advance for groups—hotel can assist. Policies can change, so confirm directly with the Jardim Botânico.

5. Buying Essentials Early/Late: - Pharmacies (Drogasil, Pacheco, Raia) in Ipanema/Jardim Botânico open 7am–10pm. Zona Sul supermarket opens 7am–10pm. Beach kiosks open by 8am, close late evening. Most markets and tourist areas have at least one early/late option.

6. Traffic Between Garden & Markets: - Morning/lunch: 20–30 min to Ipanema, up to 1 hour to São Cristóvão (worse 4–6 pm weekdays). Budget extra time for traffic, especially Fri–Sun. For São Cristóvão, consider going before 3pm or after 7pm to avoid peak jams.

7. Market Seating/Reservations: - Most market food tents (Hippie Fair, São Cristóvão) are first-come, first-served; no reservations. Arrive early for group seating. Larger restaurants inside São Cristóvão may accept reservations by phone (often only in Portuguese).

8. Group Flow at Shops/Markets: - Six is a large group for crowded stalls. Suggest splitting into pairs/trios with a set time/place to regroup (e.g., 30 min shopping, meet at juice bar or main entrance). Share your locations via WhatsApp.

9. Food Tour/Workshop Safety & Language: - Stick to highly rated providers (Eat Rio, Rio4Fun, Choco.Rio) with reviews on TripAdvisor or GetYourGuide. Confirm English language and group size when booking. For busy periods, book 4–7 days ahead.

10. Lunch Reservations & Location: - Backup restaurants (e.g., Delírio Tropical, Garota de Ipanema, Galeto Sat's) accept reservations for groups, especially on weekdays. All are within 10–15 min of main sights. Call or use Google/Instagram to reserve.

11. Beach/Park Safety Late Afternoon/Evening: - Main beaches (Ipanema, Copacabana) and parks (Parque Lage) are safe until sunset, especially in a group. Avoid isolated areas after dark, and keep valuables hidden. Stay near lifeguard posts or kiosks with staff.

12. Arpoador Facilities & Crowds: - Public toilets and refreshment stands available (small fee for toilets). Sunset is very popular—arrive 1+ hour early for a good spot, or consider nearby Ipanema Posto 8 if overcrowded. Beware of pickpockets in crowds.

13. Samba Class Booking: - Most studios (e.g., Rio Samba Dancer, Casa de Dança Carlinhos de Jesus) offer flexible scheduling for tourists, but for a group and English instruction, book at least 48 hours ahead. Some offer private group classes.

14. Lost Person Protocol: - Pre-arrange a meeting point at each location. Keep phones charged and share live location in WhatsApp. If someone is lost and unreachable, contact the venue's staff/security, or call Rio's tourist police (Dial 190 for police, 2253-1177 for tourist assistance, English spoken).

15. Inclusive Group Decisions: - Start each day with a brief meeting to gather everyone's priorities and energy levels. Use a poll (WhatsApp or verbally) for major decisions. Pair up those with similar interests if splitting up. Regroup for group meals and check-ins.

16. Handling Disagreements/Pacing: - Build in regular breaks and "optional" periods for rest or solo exploring. Rotate leadership or decision-making. If there's a dispute, defer to a coin toss or majority vote, and keep the mood light—remind everyone the goal is shared enjoyment, not rigid schedules.

---

**Host:** **Day 4: Jardim Botânico & Markets (Group Day – Six People)**

**Group Logistics & Transport** - Arrange group-friendly transport: Book an UberXL, two Ubers/taxis, or a private van (your hotel can assist). Confirm a morning meeting point and time. Allow 20–30 extra minutes for traffic, especially midday. - Create a WhatsApp group for coordination, share live locations, and agree on emergency meeting points at each stop.

**Morning: Jardim Botânico** - Arrive early (opens 8 AM) to enjoy cooler weather and avoid crowds. - Buy tickets together for possible group discount (check requirements on the official website; bring ID or passport copies). - Book an English-speaking guide in advance (via the Jardim's website or your hotel) for a more engaging visit. - The gardens are mostly accessible, but there's moderate walking—remind everyone to bring water, hats, and sunscreen (available at nearby pharmacies if needed).

**Late Morning: Market Visit** - **Sunday:** Head to the Feira Hippie de Ipanema (15 min drive), ideal for groups with spacious stalls, food, and shopping. Arrive early for best seating at

food tents. - **Weekdays:** Consider Feira de São Cristóvão (30–45 min drive), a lively, covered Northeastern Brazilian market with group-friendly restaurants (e.g., Barraca da Chiquita). For a smaller, upscale vibe, try artisanal shops in Leblon or the Ipanema Farmers' Market (Fridays). - If interested in food experiences, book a group food tour or chocolate tasting workshop (Eat Rio, Rio4Fun, Choco.Rio). Reserve 48–72 hours ahead and confirm English-speaking guides.

**Lunch** - Reserve a table for six in advance. Top options: - **Zazá Bistro** (Ipanema; excellent for dietary needs) - **Da Bela** (near Jardim Botânico; farm-to-table) - **Fogo de Chão** (Botafogo; festive churrascaria) - **Alternatives:** Delírio Tropical, Garota de Ipanema, Galeto Sat's (all group-friendly, call ahead) - Confirm allergy accommodations and dietary restrictions (collect info via WhatsApp or a simple Google Form before the day).

**Afternoon: Downtime & Social Time** - After lunch, enjoy downtime together: hotel pool, a relaxed walk on the Ipanema or Copacabana boardwalk, or a group spa session (many hotels offer packages). - For budget stays, relax at Parque Lage's café or a local juice bar (Bibi Sucos, Polis Sucos). - For sunset, head to Arpoador (arrive by 5 PM for space; safe in groups, but watch valuables and avoid lingering after dark).

**Evening (Optional)** - If energy allows, book a private group samba class (e.g., Rio Samba Dancer, Casa de Dança Carlinhos de Jesus; reserve 48 hours ahead). - Or, unwind at a casual bar or rooftop in Ipanema or Leblon (reserve for a group this size).

**Group Safety & Coordination** - Stay together in crowded markets; split into pairs/trios if needed, with set regrouping times/locations. - Carry only essentials (crossbody bags, no passports), keep phones secure, and bring small bills for markets/tips. - Review emergency meeting points at each stop. - Share live locations in WhatsApp and agree on what to do if someone gets separated (meet at entrance, contact hotel/tourist police if needed).

**Flexibility & Engagement** - Build in 1–2 hours of free time for solo exploring or rest. - Rotate who leads the day or chooses lunch. - Start each day with a quick group check-in on energy, interests, and must-sees. - If interests diverge, split for short periods and regroup for meals or main activities.

—

**Summary of Adjustments for Day 4:** - All activities are now optimized for a group of six: transport, tours, meals, downtime, and safety. - Emphasis on advance reservations for guides, tours, and restaurants. - Market and food options tailored for group tastes and dietary needs. - Practical tips for group safety, communication, and flexibility. - Built-in options for splitting up or relaxing together as needed.

