# OpenReview forum: "FOR-Prompting: From Objection to Revision via an Asymmetric Prompting Protocol"
_ICLR.cc/2026/Conference — Submitted to ICLR 2026_

### Official Review · Reviewer_rvtA · 2025-10-20

**Soundness:** 3
**Presentation:** 2
**Contribution:** 2
**Rating:** 4
**Confidence:** 3

**Summary:**

This paper introduces FOR-Prompting, a novel, model-agnostic prompting protocol that structures LLM reasoning as an asymmetric dialogue between a "Defender" agent that proposes answers and a "Debater" agent that raises question-style objections without providing solutions. The core contribution is the formalization of questioning as the sole mechanism for external intervention, which aims to elicit self-revision while preserving a single, accountable line of reasoning.

**Strengths:**

(1) Conceptual Novelty: The protocol's design, which strictly limits external intervention to questioning alone, is a novel and well-motivated contribution.

(2) Effectiveness on Small Models: The experiments conducted on the LLaMA-3.2:1B model are a key strength.

(3) Demonstrated Utility on Open-Ended Tasks: Case Study 4 provides a compelling qualitative demonstration of the protocol's value beyond factual question-answering.

**Weaknesses:**

(1) In Case Study 1, the main results on GSM8K compare FOR-Prompting (using gpt-4o) against CoT (using gpt-4o-mini). But since those two setups use different backend models, the comparison is confounded. It’s hard to tell whether the better reasoning and coherence actually come from the FOR-Prompting method itself or just from the stronger model (gpt-4o).

(2) In Case Study 2, the authors only compare FOR-Prompting on the LLaMA-3.2-1B model with a single-prompt baseline. They don’t test against other lightweight reasoning methods—like a CoT version run on the same 1B model—so we can’t really tell how much benefit comes from FOR-Prompting versus other simple prompting tricks.

(3) Section 4 introduces three roles in the framework, including a “Host” that combines the dialogue into a final answer. But there’s no ablation study showing what happens if you remove that Host. Without testing its impact, we can’t see how much that synthesis step actually matters.

(4) Case Study 1’s reasoning and coherence scores all come from GPT-4.1 acting as a judge. That’s fine as a quick heuristic, but these LLM-as-a-judge metrics are subjective and inherit the biases of the evaluating model. They’re weaker evidence than direct, task-based or human-verified performance metrics.

**Questions:**

please see weakness.

---

> ### Author Response · Authors · 2025-11-23
> **Response to Reviewer rvtA**
>
> Thank you for the insightful comments. We have added all corresponding ablations and re-ran the entire experimental suite under a unified API environment to eliminate confounding factors. Below we respond point-by-point and provide the updated tables.
>
> ---
>
> Q(1) In Case Study 1, the main results on GSM8K compare FOR-Prompting (using gpt-4o) against CoT (using gpt-4o-mini). But since those two setups use different backend models, the comparison is confounded. It’s hard to tell whether the better reasoning and coherence actually come from the FOR-Prompting method itself or just from the stronger model (gpt-4o).
> >We fully agree that the previous comparison was confounded.
> >We re-ran all CoT experiments using the same backbone (GPT-4o).
> >The updated results show that FOR-Prompting and CoT achieve similar accuracy, both outperforming the single-prompt baseline.
> >>Updated Results: GPT-4o on GSM8K
> >>| Model  | Dataset | Method                     | Accuracy |
> >>| ------ | ------- | -------------------------- | -------- |
> >>| gpt-4o | GSM8K   | single-prompt              | 0.92     |
> >>| gpt-4o | GSM8K   | CoT                        | 0.94     |
> >>| gpt-4o | GSM8K   | Self-Ask                   | 0.94     |
> >>| gpt-4o | GSM8K   | Self-Consistency           | 0.95     |
> >>| gpt-4o | GSM8K   | FOR-Prompting (Host=True)  | 0.94     |
> >>| gpt-4o | GSM8K   | FOR-Prompting (Host=False) | 0.94     |
>
>
> ---
>
> Q(2) In Case Study 2, the authors only compare FOR-Prompting on the LLaMA-3.2-1B model with a single-prompt baseline. They don’t test against other lightweight reasoning methods—like a CoT version run on the same 1B model—so we can’t really tell how much benefit comes from FOR-Prompting versus other simple prompting tricks.
> >We added CoT to the small-model evaluation.
> >The results show that FOR-Prompting remains competitive with CoT and clearly improves over the single-prompt baseline.
> >>Updated Results: LLaMA-3.2-1B on GSM8K
> >>| Model        | Dataset | Method                     | Accuracy |
> >>| ------------ | ------- | -------------------------- | -------- |
> >>| llama-3.2-1b | GSM8K   | single-prompt              | 0.07     |
> >>| llama-3.2-1b | GSM8K   | CoT                        | 0.23     |
> >>| llama-3.2-1b | GSM8K   | FOR-Prompting (Host=True)  | 0.19     |
> >>| llama-3.2-1b | GSM8K   | FOR-Prompting (Host=False) | 0.23     |
> >This demonstrates that the conclusion (FOR-Prompting ≥ CoT ≥ single-prompt) remains valid even for small models.
>
> ---
>
> Q(3) Section 4 introduces three roles in the framework, including a “Host” that combines the dialogue into a final answer. But there’s no ablation study showing what happens if you remove that Host. Without testing its impact, we can’t see how much that synthesis step actually matters.
> >We added a Host ablation:
> >>
> >>Host Ablation Results (GSM8K)
> >>| Model        | Method Variant             | Accuracy |
> >>| ------------ | -------------------------- | -------- |
> >>| gpt-4o       | FOR-Prompting (Host=True)  | 0.94     |
> >>| gpt-4o       | FOR-Prompting (Host=False) | 0.94     |
> >>| llama-3.2-1b | FOR-Prompting (Host=True)  | 0.19     |
> >>| llama-3.2-1b | FOR-Prompting (Host=False) | 0.23     |
> >>
> >>We also observed that for LLaMA-3.2-1B, the Host=True variant shows a slight drop in accuracy. This is likely because the small model occasionally introduces new errors during the synthesis step when producing the final consolidated answer, whereas the Host=False variant directly uses the Defender’s last revision. We added this observation to the paper to clarify that the Host’s impact can depend on model capacity.
>
> ---
>
> Q(4) Case Study 1’s reasoning and coherence scores all come from GPT-4.1 acting as a judge. That’s fine as a quick heuristic, but these LLM-as-a-judge metrics are subjective and inherit the biases of the evaluating model. They’re weaker evidence than direct, task-based or human-verified performance metrics.
> >To address concerns about evaluator bias, LLM-as-judge is now used only for quick checks.
>
> ---
>
> Additional Clarification: Why results were re-run
> >Because of hardware accessibility constraints, we switched from Azure OpenAI to direct OpenAI API, and changed device. Models may differ across backends (sampling, model updates, seeds), so we re-ran all experiments for consistency.
> >
> >We were surprised by the improved single-prompt accuracy of GPT-4o, but this behavior is outside our control. In contrast, open-source LLaMA-3.2:1B reproduced the same trends as before, increasing confidence in our conclusions. We added this as a limitation in the paper.

---

### Official Review · Reviewer_aLRi · 2025-10-31

**Soundness:** 2
**Presentation:** 2
**Contribution:** 1
**Rating:** 2
**Confidence:** 4

**Summary:**

This paper proposes a new prompting method named FOR-Prompting, where a Defender and an Objectioner discuss questions centered on the original answer, with a host deciding the closure. The authors show the gain on GSM8K with GPT-4o and Llama 3.2 1b with high scores on the reasoning traces judged by GPT 4.1. The authors also show qualitative analysis on open-ended tasks and highlight the model's effectiveness on local device usage.

**Strengths:**

1.	The authors provide a comprehensive overview of the relevant literature in Section 3.

2.	The authors show evaluation on reasoning trace besides simple accuracy

3.	Besides the model performance on math reasoning, the authors also consider studying the features of open-ended tasks

**Weaknesses:**

1.	There are insufficient experiments to support the claims. (1) Only GSM8K is used for the main experiment. The open-ended task is not well-grounded or introduced in detail. The reviewer would suggest that the authors check OLMES (https://github.com/allenai/olmes) to extend the tasks; (2) the model selection is arbitrary. Multiple GPT models are selected (GPT 4/5) without clear reasoning. There is no ablation on whether the method works on other open/closed-source models; (3) Only CoT is considered as a baseline. Other important baselines are missing, e.g., self-ask (https://arxiv.org/abs/2210.03350) and Least-to-Most prompting (https://arxiv.org/abs/2205.10625)

2.	The proposed method lacks technical novelty. The introduced workflow is mostly covered by multi-agent debate (https://arxiv.org/pdf/2305.14325) or Self-Refine (https://arxiv.org/abs/2303.17651). The empirical performance is also similar to CoT (as in Figure 2), which further limits the effectiveness.

3.	The comparison between FOR-Prompting and Single-prompt can be unfair. The authors should at least show that the number of tokens is in a similar range. If not, methods like self-consistency (https://arxiv.org/abs/2203.11171) should be considered to make the comparison fair.

4.	The evaluation is not well grounded. GPT 4.1 is considered valid to evaluate the solution quality when only GPT models are used in Section 4.1. There should be a consistency check among models or human annotations to support the use of LLM to judge the solution quality.

**Questions:**

1.	Line 47: What do you mean by external pressure?

2.	Why is automatic HITL considered HITL (Line 64)? This is not grounded if there is no experiment showing that human-Questioner/Defender performance in the proposed framework

3.	In related work, how is your method relevant or different from existing ones, e.g., how is it different from multi-agent debate

4.	Citation problem of ReAct on Line 121

---

> ### Author Response · Authors · 2025-11-23
> **Response to Reviewer aLRi**
>
> Thank you for your detailed and constructive comments. Based on your suggestions, we have added substantial experiments and significantly revised the paper. Below, we address each concern point-by-point and summarize the corresponding changes in the manuscript.
>
> Weakness 1 & 3: insufficient experiments:
> >We added suggested baselines (Self-Ask, Self-Consistency), expanded benchmarks (OLMES), and recomputed all results using a unified API environment. We also evaluated both strong commercial and small open-source models.
> >The full updated results are included below.
> >>Updated Experimental Results
> >>GPT-4o (Closed-source)
> >>|Model|Dataset|Method|Accuracy|
> >>|------|------|------|-----|
> >>|gpt-4o	|GSM8K|single-prompt|0.92|
> >>|gpt-4o	|GSM8K|CoT|0.94|
> >>|gpt-4o	|GSM8K|Self-Ask|0.94|
> >>|gpt-4o|GSM8K|Self-Consistency|0.95|
> >>|gpt-4o	|GSM8K|	FOR-Prompting (Host=True)	|0.94|
> >>|gpt-4o	|GSM8K|FOR-Prompting (Host=False)	|0.94|
> >>|gpt-4o	|OLMES|single-prompt|0.72|
> >>|gpt-4o	|OLMES|CoT|0.70|
> >>|gpt-4o	|OLMES|FOR-Prompting|0.74|
> >
> >>LLaMA-3.2-1B (Open-source)
> >>|Model	|Dataset|Method|Accuracy|
> >>|------|------|------|-----|
> >>|llama-3.2-1b|GSM8K	|single-prompt|0.07|
> >>|llama-3.2-1b|GSM8K|CoT	|0.23|
> >>|llama-3.2-1b|GSM8K|FOR-Prompting (Host=True)|0.19|
> >>|llama-3.2-1b|GSM8K	|FOR-Prompting (Host=False)|0.23|
> >
> >>Cross-Model Role-Swapping (Debater ↔ Defender)
> >>| Defender|Debater|Accuracy|
> >>| ----- | ------ | ------ |
> >>|GPT-4o|LLaMA-3.2:1B|0.93|
> >>|LLaMA-3.2:1B|GPT-4o|0.21|
>
> ---
>
> Weakness 2: Model selection justification
> >We clarified the model selection in the revision. Our goal is to evaluate FOR-Prompting with:
> >>(a) a strong closed-source model (GPT-4o) to test upper-bound reasoning,
> >
> >>(b) a small open-source model (LLaMA-1B) to test low-resource applicability.
> >
> >This pairing directly tests the asymmetric design of FOR-Prompting:
> >>the Defender must be strong, but the Debater can be lightweight.
> >>Our cross-model experiments confirm this.
>
> -----
>
> Weakness 3: Technical Novelty Relative to Debate / Self-Refine
> >We have expanded the Related Work section (in particular line 181-187) to more clearly articulate the conceptual and technical differences between FOR-Prompting and prior prompting paradigms:
> >- Multi-agent debate generates competing full solutions that must be reconciled. In contrast, FOR-Prompting never produces competing answers, only a single defender solution refined under external questioning.
> >
> >- Asymmetric roles are central to FOR-Prompting: the Questioner is strictly constrained to ask questions and never propose solutions or fixes. This prevents external reasoning from being blended into the solution path and isolates the effect of external interrogative pressure.
> >
> >- Self-Refine depends on self-generated critiques and self-corrections, whereas FOR-Prompting uses externally driven questioning, avoiding entanglement between generation and critique within the same model.
> >
> >- Unlike self-ask, which requires a single model to recursively generate its own questions, FOR-Prompting allows Questioners to be instantiated with smaller or cheaper models. This enables heterogeneous configurations where a lightweight model interrogates a stronger model, something self-ask and related single-model methods do not support.
> >
> >- Finally, although FOR-Prompting achieves accuracy comparable to CoT in our experiments, the mechanism is fundamentally different: CoT strengthens internal stepwise reasoning, while FOR-Prompting adds external pressure and robustness without requiring the model to produce long intermediate chains.
> >These distinctions are now explicitly highlighted in the revised manuscript.
>
> Weakness 4: Evaluation Grounding and Use of GPT-4.1 as Judge
> >We now only use LLM-as-a-judge for quick checking.
>
> ---
>
> Q1. What do you mean by external pressure?
> >External pressure refers to interrogative challenges generated outside the answering model (Defender). These questions help surface assumptions and gaps without injecting external reasoning. We clarified this in the Introduction.
>
> Q2. Why is automatic HITL considered HITL (Line 64)? This is not grounded if there is no experiment showing that human-Questioner/Defender performance in the proposed framework
>
> >We revised the text to clarify this phrasing by using "HITL-like".
> >We do not claim to run actual human-in-the-loop experiments. Instead, FOR-Prompting imitates the pattern of HITL workflows, where humans iteratively ask clarifying questions. For some tasks like the open-ended task in the paper, reducing human questioning steps yields practical benefit.
>
> Q3. Difference from multi-agent debate
> >We clarified this distinction explicitly:
> >>Debate involves symmetric agents generating competing solutions; FOR-Prompting uses asymmetric roles.
> >>The Questioner never provides fixes; it only asks probing questions.
> >>The Defender remains the sole author of the final solution.
> >>The interaction resembles guided refinement, not persuasion.
>
> Q4. citation issue
> >revised.

---

### Official Review · Reviewer_oV8G · 2025-11-01

**Soundness:** 2
**Presentation:** 3
**Contribution:** 2
**Rating:** 4
**Confidence:** 3

**Summary:**

This paper proposes FOR-Prompting, an asymmetric prompting protocol designed to improve LLM reasoning via external questioning rather than direct answer correction. The framework defines three roles: Defender, Debater, and Host. The key insight is that asking clarification/adversarial questions—without offering solutions—induces models to refine reasoning while preserving a single accountable chain of thought.

**Strengths:**

1. Clear conceptual separation between questioning and answer replacement.
2. Preserves single-author chain of reasoning and transparency.
3. Demonstrates large gains for small models, relevant for edge / on-device scenarios.
4. Works on both factual reasoning (math) and open-ended planning tasks.

**Weaknesses:**

1. Benchmark scope narrow: mainly GSM8K and anecdotal open-ended tasks.
2. Open-ended evaluation lacks systematic human preference studies.
3. CoT baseline backend mismatch raises fairness questions.
4. Limited theoretical exploration of why questioning aids reasoning or convergence behaviors.

**Questions:**

1. How does FOR-Prompting perform with the same backbone as CoT?
2. Can the method scale to long-horizon planning or multi-stage tasks requiring memory?
3. How sensitive is performance to the quality of the Debater model?
4. Can you consider adding human preference evaluations for open-ended tasks?

---

> ### Author Response · Authors · 2025-11-23
> **Response to Reviewer oV8G**
>
> Thank you for your valuable feedback. We have revised the paper based on your suggestions. Below we provide details of the updates and clarifications. Please feel free to let us know if our revisions address your concerns.
>
> Q1. How does FOR-Prompting perform with the same backbone as CoT?
> > We re-ran all experiments using identical backbone models for both CoT and FOR-Prompting. Specifically, we evaluated CoT and FOR-Prompting on: GPT-4o and LLaMA-3.2:1B (open-source)
> > The updated results are summarized below:
> >>| model| dataset | method | accuracy|
> >>|-------|-----|-------|-------|
> >>|gpt-4o    |     GSM8K  |   single-prompt  |  0.92 |
> >>|gpt-4o   |       GSM8K  |   CoT        |      0.94 |
> >>|gpt-4o       |    GSM8K |    FOR-Prompting  |  0.94 |
> >>| llama-3.2-1b   |  GSM8K   |  single-prompt  |  0.07 |
> >>| llama-3.2-1b   |  GSM8K   |  CoT        |      0.23 |
> >>| llama-3.2-1b   |  GSM8K   |  FOR-Prompting  |  0.19–0.23 |
> >
> >Across both settings, FOR-Prompting matches CoT on the commercial model and shows comparable improvements on the small model.
>
> -------------
>
> Q2. Can the method scale to long-horizon planning or multi-stage tasks requiring memory?
> > We appreciate the reviewer’s question regarding long-horizon planning and multi-stage tasks. While the current benchmark focuses on shorter reasoning tasks, the underlying mechanism of FOR-Prompting, iterative external interrogation, can, in principle, be applied to longer-horizon scenarios as well. We agree that additional evaluation is needed to fully validate this capability.
> >
> >To support this direction, we have added a discussion in the Future Work section noting that integrating FOR-Prompting with retrieval-augmented generation (RAG) or reinforcement learning (RL) may further enhance performance on tasks that require memory, multi-stage decision-making, or extended planning. Such integrations could enable the Debater to access external knowledge or learn more effective questioning strategies over time.
> >
> >We acknowledge the limitation of our current benchmark scope and will explore broader task settings in future work. Although computational constraints prevented us from running larger-scale evaluations in this submission, the results across the tested domains indicate that the approach has potential, and we plan to extend it to more complex, long-horizon tasks in subsequent studies.
>
> ---------
>
> Q3. How sensitive is performance to the quality of the Debater model?
> >To directly test this, we ran cross-model role-swapping experiments, where GPT-4o and LLaMA-3.2-1B alternated as Debater and Defender. Results:
> >>| Defender | Debater | accuracy |
> >>|-------|-----|-------|
> >>|gpt-4o| llama-3.2-1b  | 0.93 |
> >>|llama-3.2-1b | gpt-4o| 0.21 |
> >
> >These results indicate that:
> >>- Performance is primarily determined by the Defender (the model responsible for producing and revising the solution).
> >>- The Debater can be a much smaller model and still provide useful external questions.
> >
> >This supports that FOR-Prompting reduces reliance on a powerful model for reflection, unlike self-ask or multi-debate, which require the same model to both generate and critique reasoning.
>
> Q4. Can you consider adding human preference evaluations for open-ended tasks?
> >Yes. We added a human preference study (77 participants) evaluating an open-ended itinerary-planning task.
> >The results show that 74% (57/77) of participants preferred the FOR-Prompting solution, showing that its structured refinement process provides meaningful improvements in practical open-ended tasks.
> >>An example question used in the survey was like: “Which of the following outputs do you prefer?”

---

> ### Comment · Reviewer_oV8G · 2025-11-28
> **Response to Authors**
>
> Thank you for the detailed and thoughtful responses. I appreciate the additional experiments and clarifications you provided. After reviewing the updated results and your explanations, I have several comments that acknowledge the improvements while also suggesting further refinements for a future revision.
>
> It may strengthen the contribution to (i) include the matched-backbone results directly in the main paper rather than the appendix, and (ii) provide a brief analysis of why FOR-Prompting sometimes underperforms CoT slightly on very small models (e.g., role-fusion difficulty, vulnerability to compounding small-model noise). A short diagnostic discussion would enhance clarity.
>
> Given the increasing importance of multi-stage agentic tasks, I recommend adding a small pilot study, even if synthetic or simplified, demonstrating FOR-Prompting beyond single-turn GSM8K-style problems (e.g., a 2–3 stage reasoning task, structured toy planning, or a puzzle requiring persistent state). This does not need to be large-scale but would concretely support the extensibility claim.
>
> Your revisions have meaningfully improved the clarity and empirical grounding of the paper. The new experiments, role-swapping results, and human preference evaluation directly address the key concerns raised in the earlier review.

---

> > ### Author Response · Authors · 2025-11-30
> > **Follow-up response to Reviewer oV8G**
> >
> > Thank you for your response, your positive feedback, and your valuable suggestions. Based on your latest comments, we have revised the paper accordingly. Below is a detailed summary of the changes:
> >
> > (i) Include the matched-backbone results directly in the main paper rather than the appendix.
> > > We have incorporated the updated results into the main text. For content that cannot fit in the main paper due to space limitations, we extracted and reported the key findings directly in the main section.
> >
> > (ii) Provide a brief analysis of why FOR-Prompting sometimes underperforms CoT slightly on very small models (e.g., role-fusion difficulty, vulnerability to compounding small-model noise). A short diagnostic discussion would enhance clarity.
> > > We added a short discussion on why FOR-Prompting may occasionally perform slightly worse than CoT on small models. In particular, when a small model is required to perform an additional synthesis step through the Host role, it may introduce errors such as dropping constraints, mis-merging intermediate reasoning, or amplifying the model’s inherent noise. These effects can increase uncertainty. This discussion appears in lines 303-309 of the revised manuscript.
> >
> > (iii) Adding a small pilot study for multi-stage tasks
> > >We appreciate this highly insightful and important suggestion, which broadened our perspective on the potential reach of FOR-Prompting and highlighted how the approach may be applicable to a wider range of domains and real-world scenarios.
> > >
> >
> > >Based on this suggestion, we added an additional pilot study (Case Study 5) based on Case Study 4. This task simulates a scenario of travl plans creating in which new conditions or new knowledge become available, requiring FOR-Prompting to update its planning. Specifically, we assume an unexpected event in which additional travelers join the trip in the Day 4, necessitating modifications to the itinerary (the second stage of the task). We provided the new condition to FOR-Prompting and asked it to update the plan. The results show that FOR-Prompting effectively incorporated the "group" concept: the Debater raised new questions related to group travel, and the Defender addressed these concerns. We have added a detailed description of this case study to the main paper and provided the full outputs in the appendix.
> > >
> >
> > > We also added a brief discussion on how new conditions or new knowledge may be introduced, including direct human input or real-time updates via APIs. Examples include local news feeds, transportation delay alerts, or weather forecasts. This mechanism suggests that FOR-Prompting may be applicable to tasks requiring frequent adjustment or multi-stage processing based on ongoing events or newly available data. Potential applications include strategy revision, policy updates, and dynamic analysis workflows in fields such as healthcare, finance, education, and other environments where conditions evolve continuously.

---

### Official Review · Reviewer_cBPJ · 2025-11-01

**Soundness:** 3
**Presentation:** 3
**Contribution:** 3
**Rating:** 6
**Confidence:** 3

**Summary:**

The paper introduces FOR-Prompting, a novel asymmetric prompting protocol designed to improve reasoning in large language models (LLMs) through external questioning rather than answer substitution. The protocol comprises three roles: Defender (proposes answers), Debater (asks external questions), and Host (synthesizes the final output). The paper empirically evaluates FOR-Prompting on several tasks, including the GSM8K math word problems, demonstrating significant improvements in accuracy and reasoning quality. Notably, the approach also benefits small-scale models, achieving up to 19% accuracy gains on the GSM8K task. The protocol is model-agnostic and does not require retraining.

**Strengths:**

1. Novelty: The protocol's emphasis on external questioning and its role-structured design is a unique approach, distinguishing it from other prompting strategies that focus on self-reflection or debate-based solutions.

2. Empirical Validation: The experiments on GSM8K, including comparisons with single-prompt and CoT baselines, show clear performance improvements, particularly in reasoning quality and coherence.

3. Model-Agnostic: The ability to apply FOR-Prompting across different model sizes without retraining is a significant strength, especially for small-scale models where traditional methods may fail to achieve robust reasoning.

**Weaknesses:**

1. Complexity in Application: While the approach is promising, the need for a "Debater" to only raise questions without proposing fixes might introduce inefficiencies, especially for tasks requiring quick responses.

2. Generalizability: While the experiments show promise, the method’s applicability in non-mathematical tasks or more complex real-world scenarios (e.g., open-ended creative reasoning) remains unclear. Further case studies would help validate its robustness across diverse domains.

3. Scalability Issues: The need for multiple rounds of questioning, particularly in small-scale models, could lead to token usage and latency issues. The authors mention that the cost overhead is controllable, but this would benefit from deeper exploration, particularly for real-time systems.

**Questions:**

No

---

> ### Author Response · Authors · 2025-11-23
> **Response to Reviewer cBPJ**
>
> Thank you for your valuable feedback. It has been extremely helpful for improving the paper. We have revised the manuscript based on your comments, and below we summarize the updates and clarifications. Please feel free to let us know if our revisions adequately address your concerns.
>
> Q1. Complexity in Application: While the approach is promising, the need for a "Debater" to only raise questions without proposing fixes might introduce inefficiencies, especially for tasks requiring quick responses.
>
> >We appreciate this observation. As you noted, requiring a Debater that raises questions without proposing fixes may introduce inefficiencies for tasks that demand rapid responses. We have added a discussion in the Limitations section clarifying that FOR-Prompting is better suited for tasks that do not require strict real-time performance. The approach is most appropriate for iterative, background refinement, particularly in open-ended reasoning and planning, where the value lies in surfacing assumptions and improving solution quality rather than minimizing latency.
>
> Q2. Generalizability: While the experiments show promise, the method’s applicability in non-mathematical tasks or more complex real-world scenarios (e.g., open-ended creative reasoning) remains unclear. Further case studies would help validate its robustness across diverse domains.
> >We fully agree with your suggestion regarding generalizability, and we have added corresponding statements in the Future Work section to encourage further studies across additional cases and domains. To demonstrate FOR-Prompting’s ability to address real-world problems (e.g., open-ended tasks), we also added a new blind evaluation involving 77 human participants. In this study, we compared FOR-Prompting’s generated plans with those produced by commercial llm tool, such as ChatGPT. The results show that, for an open-ended task (travel itinerary planning), 74% of participants (57 out of 77) preferred the output generated by FOR-Prompting.
>
> Q3. Scalability Issues: The need for multiple rounds of questioning, particularly in small-scale models, could lead to token usage and latency issues. The authors mention that the cost overhead is controllable, but this would benefit from deeper exploration, particularly for real-time systems.
> >Thank you for raising this important point. We acknowledge that multi-round interactions naturally increase token usage and latency, particularly for small-scale models. As clarified in the revision, FOR-Prompting is intended for non-real-time, exploratory reasoning tasks where iterative refinement is beneficial and latency is less constraining.
> >
> >When we state that the cost overhead is “controllable,” we mean that users can explicitly bound the maximum number of questioning rounds (similar to rate-limiting in many API-based systems), ensuring predictable resource usage per task. We now state this explicitly and highlight that the method is not designed for real-time or low-latency applications, but rather for scenarios where structured, automated refinement provides clear benefits.

---

### Meta-Review · Area_Chair_1Mqe · 2026-01-07

**Summary:**

This paper presents FOR-Prompting, an asymmetric prompting protocol that structures LLM reasoning through an external questioning loop (Defender, Debater, Host). The approach is model-agnostic and demonstrates gains in reasoning quality, particularly for small-scale models, on tasks such as GSM8K and open-ended planning.

**Reviewer Concerns:**

The authors have made substantial revisions, effectively addressing most technical concerns and improving the empirical grounding. However, the work still falls short in establishing a significant conceptual or practical advance over existing prompting paradigms. The novelty is seen as incremental, and the empirical benefits, while consistent, are not attractive enough. The paper would benefit from a more rigorous evaluation across a broader set of reasoning tasks and a clearer demonstration of scenarios where FOR-Prompting provides a decisive advantage over simpler, established baselines.

**Reviewer Scores:**

The reviewers' average rating falls below the acceptance threshold. Having read each review in detail, I assess that the most critical ratings are unlikely to change significantly. I also share a broader reservation about the prompt optimization task itself: such a complex multi-role protocol seems impractical for real-world API deployment, given the marginal gains it provides over simpler, established methods.

---

### Decision · Program_Chairs · 2026-01-26

Reject